# TEST-TIME OPTIMIZATION OF 3D POINT CLOUD LLM VIA MANIFOLD-AWARE IN-CONTEXT GUIDANCE AND REFINEMENT

**Tiankai Chen**[1,3], **Nanqing Liu**[2], **Li Yang**[1], **Xulei Yang**[3], **Tianrui Li**[1], **Xun Xu**†[3]

[1]Southwest Jiaotong University, [2]Yunnan Normal University
[3]Institute for Infocomm Research (I²R), A*STAR
{chentiankai, yangli_ef}@my.swjtu.edu.cn, lansing163@163.com,
{yang_xulei, xu_xun}@a-star.edu.sg, trli@swjtu.edu.cn

## ABSTRACT

Multimodal Large Language Models (MLLMs) have demonstrated impressive capabilities in textual and 2D visual reasoning, yet their ability to understand and reason over 3D data remains limited. The issues become more challenging for understanding standalone 3D point cloud due to the high interclass confusion. In this work, we propose **P**oint-**G**raph **LLM** (PGLLM), a framework that enables more effective 3D point cloud understanding by integrating in-context prompting and score refinement at test-time, respecting supporting data manifold. Our method first employs a pre-trained point cloud encoder which are used to construct a graph where edges encode visual similarity. Each support point cloud sample is converted to a textual caption via pre-trained PointLLM. For a test query, the graph is used to retrieve relevant neighbors whose captions serve as contextual demonstrations for a second stage LLM for final reasoning, a process we term in-context guidance. Furthermore, we introduce a confidence score refinement mechanism based on label propagation to enhance the reliability of LLM predictions for classification and out-of-distribution (OOD) detection tasks. All the above optimizations are carried out fully at test-time. Extensive experiments across diverse 3D datasets and tasks demonstrate that PGLLM consistently improves accuracy and robustness over prior baselines with very almost no additional computation cost, showcasing a promising direction toward native 3D reasoning with MLLMs. The code is available on https://github.com/handsome999KK/PGLLM

## 1 INTRODUCTION

While multimodal large language models (MLLMs) have revolutionized textual and 2D visual reasoning, their ability to interpret and reason about 3D environments remains fundamentally limited. Recent efforts (Tang et al., 2024; Qi et al., 2024a) have explored 3D understanding with MLLMs by equipping them with mechanisms to perceive 3D information, often through auxiliary modalities or intermediate representations. Typically, these approaches operate by either projecting 3D point clouds into 2D images (Zhu et al., 2023) or by piping pre-extracted features into the language model (Xu et al., 2024; Guo et al., 2023). While these pipelines demonstrate promising results, they fall short in granting MLLMs direct access to the underlying geometric structure of 3D data. Consequently, enabling MLLMs to natively process and reason over rich 3D point cloud information remains an open and challenging research question.

A notable step in this direction is PointLLM (Xu et al., 2024), which introduces a framework capable of understanding colored 3D object point clouds in response to human instructions. PointLLM fuses geometric, appearance, and linguistic information by coupling a point cloud encoder with a pre-trained LLM such as LLaMA (Touvron et al., 2023). To handle downstream tasks such as classification or captioning, PointLLM adopts a two-stage pipeline, where the PointLLM first

† Correspondence to <xu_xun@i2r.a-star.edu.sg>. This work was partially done during Tiankai Chen and Nanqing Liu's visit to I²R funded by China Scholarship Council.

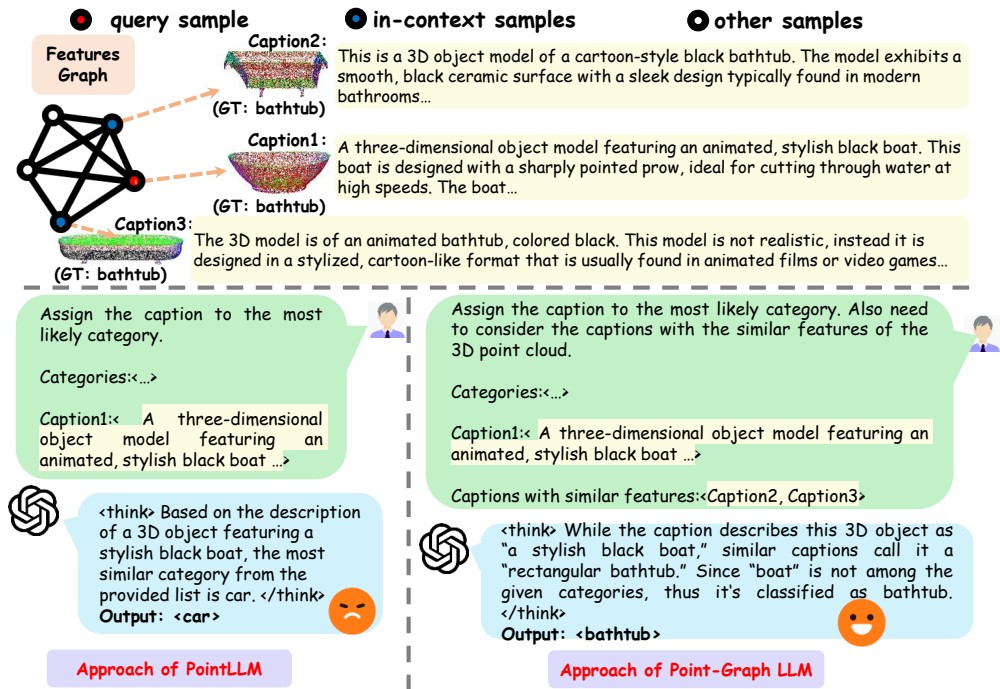

Figure 1: Manifold-aware in-context guidance leverages the 3D captions of adjacency nodes as demonstration for downstream understanding tasks.

generates a textual description, and a second stage LLM interprets this output for final task-specific predictions. While this method demonstrates competitive performance, it faces a key limitation, high inter-class visual similarity in 3D point clouds often leads to confusion, particularly when each point cloud is interpreted in isolation. As a result, models may struggle to distinguish fine-grained differences between closely related categories, leading to reduced reliability.

To overcome this limitation, we draw inspiration from the recent success of In-Context Learning (ICL) (Agarwal et al., 2024; Li et al., 2023a; Brown et al., 2020), which enables LLMs to generalize to novel tasks by conditioning on a small set of demonstrations provided in the prompt. ICL has shown strong performance across various domains, including multi-modal settings (Tsimpoukelli et al., 2021; Jiang et al., 2024b; Huang et al., 2024), where LLMs are guided by examples combining visual examples and text descriptions. The effectiveness of ICL crucially depends on the availability of informative and task-relevant demonstrations, e.g. image-text pairs. Selecting relevant and conducive demonstrations thus remain an open challenge.

To address this challenge, we propose an in-context learning strategy that leverages the manifold structure of unlabeled supporting data, which may be maintained independently or drawn from the test set. We represent this manifold as a graph, where each node corresponds to a 3D point cloud sample. To enrich each node, we employ PointLLM to generate a caption using the standard prompt "*What is this?*". The affinity between nodes is computed based on feature similarity in the embedding space of a pre-trained 3D encoder (Yu et al., 2021a). At inference time, for a given query point cloud, we identify its neighboring nodes in the graph and append their corresponding 3D captions to the query prompt. This augmented prompt is then passed to the second-stage LLM (e.g., Chat-GPT), enabling it to perform context-aware reasoning. We refer to this mechanism as in-context guidance, which injects semantically relevant knowledge into the LLM's reasoning process at test time, without requiring model retraining.

Moreover, for classification-related downstream tasks such as recognition and out-of-distribution (OOD) detection, the confidence calibration of LLM outputs is critical for robust decision-making (Xiao et al., 2022). Relying solely on raw predictions from LLMs may be risky, especially when predictions are overconfident or miscalibrated. To address this, we further tap on the manifold build upon the supporting data and employ a refinement step, where predicted confidences are used to smooth and correct noisy labels on the graph. This is implemented via a lightweight label propagation algorithm (Zhu & Ghahramani, 2002), allowing the model to refine its predictions by

considering the consistency and confidence of nearby nodes. In this way, even anecdotal or ambiguous predictions on individual samples can be improved by leveraging the collective structure of the data.

In summary, we propose a novel 3D point cloud understanding framework **P**oint-**G**raph **LLM** that integrates in-context learning, graph-based reasoning, and confidence-aware label propagation. By bridging the gap between geometric perception and LLM reasoning, our approach enables more accurate, interpretable, and robust 3D understanding. The main contributions are summarized as follows.

- We propose an in-context guidance mechanism to optimize the effectiveness of 3D point cloud LLM for downstream tasks. This approach leverages the test-time data manifold to construct helpful demonstrations to enrich the prompt.

- We introduce a score-based inference mechanism that further improves the performance of LLMs on classification related downstream tasks by refining the initial predictions on individual samples.

- We conduct extensive experiments on multiple downstream tasks and diverse 3D point cloud datasets, demonstrating consistent performance gains over existing baselines across all settings.

## 2 RELATED WORK

**Large Language Models for 3D Understanding.** Large Language Models (LLMs) have demonstrated remarkable performance across a wide range of natural language and 2D vision tasks (Brown et al., 2020; Achiam et al., 2023; Touvron et al., 2023). Recently, there has been growing interest in extending LLMs to 3D understanding (Hong et al., 2023; Qi et al., 2024a;b; Yang et al., 2025b; Xu et al., 2024; Guo et al., 2023; Yuan et al., 2025). 3D-LLM (Hong et al., 2023) introduces a family of LLMs grounded in physical 3D reasoning, laying the foundation for language-guided 3D perception. Subsequent approaches such as PointLLM (Xu et al., 2024) and Point-Bind (Guo et al., 2023) directly process colored object point clouds by combining point cloud encoders with pretrained LLMs, enabling open-vocabulary 3D understanding. ShapeLLM (Qi et al., 2024a) pioneers the use of multi-view distillation and introduces the ReCon++ encoder, establishing a 3D multimodal evaluation benchmark (3D MM-Vet) to unify embodied 3D interaction tasks. GPT4Point (Qi et al., 2024b) extends the capabilities of LLMs to handle point cloud captioning and visual question answering. Similarly, LiDAR-LLM (Yang et al., 2025b) focuses on outdoor scene understanding by integrating LiDAR data with LLMs for large-scale 3D reasoning. While these methods demonstrate the potential of LLMs for 3D tasks, they typically treat each point cloud in isolation and do not fully exploit the structure of the data manifold.

**In-Context Learning.** In-context learning (ICL) (Brown et al., 2020) enables large language models to perform downstream tasks by conditioning on a set of demonstrations, without any parameter updates. Initially developed for natural language tasks(Zhang et al., 2022; Li et al., 2023b), ICL has since been extended to multi-modal domains. For instance, Flamingo (Alayrac et al., 2022) adapts ICL to vision-language tasks by incorporating cross-modal attention, while many-shot prompting strategies (Jiang et al., 2024b; Huang et al., 2024) have been shown to significantly enhance the effectiveness of ICL in image classification and question answering. These works demonstrate that with appropriate context, LLMs can generalize across diverse tasks in a flexible and label-efficient manner. However, applying ICL to 3D understanding remains underexplored, particularly in scenarios where constructing relevant demonstrations is non-trivial due to limited supervision.

**Manifold Learning for Visual Tasks.** Manifold learning has been widely adopted to enhance visual recognition tasks by modeling the intrinsic geometry of data distributions. Graph-based methods (Li et al., 2025; Stojnić et al., 2024; Chen et al., 2025) leverage similarities in feature space to propagate high-confidence labels or scores, improving performance in zero-shot and few-shot settings. For instance, label propagation techniques have been combined with graph neural networks (Bao et al., 2024; Stadler et al., 2021; Wu et al., 2023) to improve the separation between in-distribution (ID) and out-of-distribution (OOD) samples. Beyond classification, graph-based selection has also been applied in 3D domains, as demonstrated in GraphI2Ps (Bie et al., 2025), which filters false matches

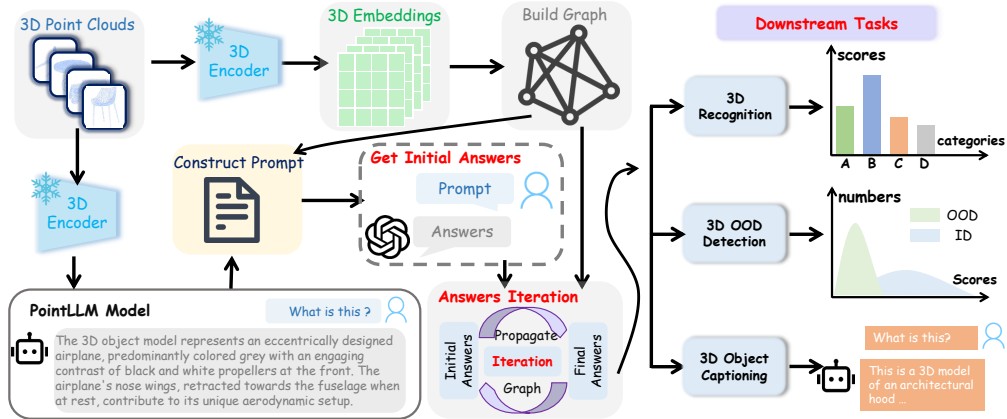

Figure 2: Overview of the proposed framework for PGLLM. After encoding the 3D test samples, the framework feeds them into PointLLM for caption generation and uses them to construct a KNN graph. Initial answers are then synthesized via LLM inference. Subsequently, leveraging relational structures within the KNN graph, we introduce an answer iteration mechanism to optimize performance on downstream tasks.

during point cloud registration via neighborhood pruning. While these methods focus on traditional backbones or vision-language models, our work is the first to integrate manifold learning into the in-context learning process of LLMs, enabling context-aware reasoning over 3D point clouds by structurally selecting and organizing prompts from the data manifold.

## 3 METHODOLOGY

### 3.1 PRELIMINARIES: POINTLLM AND 3D CAPTIONING

We begin by formalizing the setup. Let $\mathcal{D}_u = \{x_i\}_{i=1}^{N_u}$ denote an unlabeled 3D point cloud support dataset with $l$ categories, $\mathcal{D}_u$ may be the testing dataset or any unlabeled reference dataset. Let $f_p$ be a pre-trained point cloud encoder and we apply $f_p$ to the dataset yields a sequence of point cloud features:

$$\mathcal{P} = \{p_1, p_2, \ldots, p_{N_u}\}, \quad p_i = f_p(x_i).$$

For each sample $x_i$, we use the default prompt "*What is this?*" with PointLLM (Xu et al., 2024) to generate a textual caption $c_i$, resulting in a caption set, $\mathcal{C} = \{c_1, c_2, \ldots, c_{N_u}\}$. These captions serve as high-level semantic descriptions of the point clouds and form the basis for further in-context learning and downstream tasks.

### 3.2 IN-CONTEXT GUIDANCE VIA GRAPH NEIGHBOR RETRIEVAL

In-context learning (ICL) allows LLMs to perform downstream tasks by conditioning on a set of relevant demonstrations embedded within the input prompt. To construct effective in-context guidance for each test sample, we build a graph $G = (V, E)$, where each node $v_i \in V$ represents a point cloud $x_i$, and edges $e_{ij} \in E$ encode pairwise similarity between samples. Specifically, we compute the cosine similarity between point cloud features $p_i$ and $p_j$, and define symmetric the edge weight matrix $W \in \mathbb{R}^{N_u \times N_u}$ following a K-Nearest Neighbors (KNN) criterion,

$$W_{ij} = \begin{cases} e_{ij} & \text{if } e_{ij} \in \text{Top}_K(\{e_{ij}\}_{j=1}^{N_u}) \\ 0 & \text{otherwise} \end{cases}, \quad s.t. \quad e_{ij} = \frac{<p_i, p_j>}{||p_i|| \cdot ||p_j||} \tag{1}$$

For each query sample $x_i$, we retrieve its $K$ nearest neighbors on the graph, yielding a neighbor set $\mathcal{X}_i = \{x_{i_1}, \ldots, x_{i_K}\}$ and the corresponding caption set $\mathcal{C}_i = \{c_{i_1}, \ldots, c_{i_K}\}$. These captions serve as in-context demonstrations appended to the prompt, illustrated in Fig. 1 as "*Caption2*", "*Caption3*", etc. This enables the LLM to reason about the query with reference to structurally similar samples. New query samples not attached to the supporting graph can be integrated into the

graph without much computing overhead following the dynamic graph expansion scheme (Li et al., 2025).

## 3.3 SCORE REFINEMENT VIA LABEL PROPAGATION

Unlike conventional classification methods that directly predict class labels, we guide the LLM to output class confidence scores for each 3D caption. This score-based formulation enhances robustness and enables downstream tasks such as OOD detection.

**3D Recognition**: Given a 3D caption $c_i$, we prompt the LLM to output a per-category score, $S_l^{(i)} \in \mathbb{R}^l$, where each element reflects the confidence of respective class label. Aggregating across the dataset yields the initial score matrix, $S_0 \in \mathbb{R}^{l \times N_u}$. To refine predictions by leveraging geometric similarity among point clouds, we apply label propagation (Zhu & Ghahramani, 2002) over the graph $W$. Let $S_t$ denote the refined score matrix at iteration $t$. The update rule is:

$$S_t = \alpha S_{t-1} \tilde{W} + (1-\alpha) S_0, \quad \tilde{W} = D^{-\frac{1}{2}} W D^{-\frac{1}{2}}, \quad D = \text{diag}(\sum_j W_{ij}),$$

$$\hat{y}_j = \arg\max_i (S_t)_{ij}$$

where $\hat{\mathbf{y}}$ gives the final predicted class for each sample, and $\alpha$ controls the balance between the initial LLM output and propagated scores.

**3D OOD Detection**: For OOD detection, we prompt the LLM to produce a single confidence score $S(x_i) \in \mathbb{R}$ for each caption, indicating its similarity to the known in-distribution classes. A threshold $\delta$ is used to determine OOD status:

$$\hat{\mathbf{y}} = \begin{cases} \text{OOD} & \text{if } S(x_i) \leq \delta, \\ \text{ID} & \text{otherwise.} \end{cases}$$

This scalar score can be smoothed through the same graph-based propagation mechanism as in Eq. 2 with $l = 2$.

**3D Caption Refinement with In-Context Learning**: For the 3D object captioning task, we enhance the semantic quality of the initial PointLLM-generated captions using in-context refinement. We reuse the same graph-based strategy to select neighboring samples with semantically and structurally relevant captions. These are appended as demonstrations to the input prompt for each query caption. Unlike recognition tasks, the goal here is caption correction rather than classification. We guide the LLM to preserve the original semantics but improve fluency and fix any factual errors. This leverages the LLM's generative ability to produce more accurate and natural descriptions.

## 3.4 FRAMEWORK OVERVIEW

An overview of PGLLM is presented in Fig. 2. First, we extract 3D features from the test set using a frozen point cloud encoder. These features are passed to PointLLM to generate initial captions. A KNN graph is then constructed over the feature space to capture local geometric relationships. Based on this graph, we identify structurally similar neighbors for each sample, whose captions are used as in-context demonstrations. Depending on the downstream task we summarize the test-time optimization practices as follows. An illustration is deferred to the Appendix.

- For 3D recognition, the LLM outputs class-wise scores, which are further refined via graph-based label propagation.
- For OOD detection, scalar similarity scores are computed, refined via graph-based label propagation and eventually thresholded to determine OOD status.
- For captioning, in-context refinement is applied to enhance the quality of generated text while maintaining semantic fidelity.

This unified framework leverages LLMs not only as language generators but also as structured reasoning engines capable of adapting to multiple 3D tasks with minimal supervision.

| Method | one Stage LLM | 2nd Stage LLM | ModelNet40 | | | | | | | | ShapeNetCore | | | | | | | |
|---|---|---|---|---|---|---|---|---|---|---|---|---|---|---|---|---|---|---|
| | | | MN1 | | MN2 | | MN3 | | Average | | SN1 | | SN2 | | SN3 | | Average | |
| | | | AUROC↑ | FPR95↓ | AUROC↑ | FPR95↓ | AUROC↑ | FPR95↓ | AUROC↑ | FPR95↓ | AUROC↑ | FPR95↓ | AUROC↑ | FPR95↓ | AUROC↑ | FPR95↓ | AUROC↑ | FPR95↓ |
| MCM(Ming et al., 2022) | – | – | 85.3 | 53.6 | 80.2 | 74.2 | 77.5 | 72.6 | 81.0 | 66.8 | 85.1 | 51.6 | 83.2 | 46.1 | 66.4 | 75.8 | 78.2 | 57.8 |
| NegLabel(Jiang et al., 2024a) | – | – | 74.3 | 77.4 | 65.8 | 86.6 | 61.5 | 81.9 | 67.2 | 82.0 | 60.6 | 87.8 | 80.6 | 78.6 | 88.0 | 48.3 | 76.4 | 71.6 |
| ZLaP(Kalantidis et al., 2024) | – | – | 72.8 | 99.8 | 86.1 | 61.1 | 70.8 | 76.0 | 76.6 | 79.0 | 88.2 | 52.8 | 72.3 | 66.4 | 77.4 | 90.0 | 79.3 | 69.7 |
| GSP(Chen et al., 2025) | – | – | 82.4 | 77.0 | 77.9 | 65.4 | 76.1 | 76.5 | 78.8 | 73.0 | 90.6 | 38.9 | 70.7 | 64.0 | 79.7 | 93.7 | 80.4 | 65.5 |
| PointLLM(Xu et al., 2024) | PointLLM-7B | GPT-4 | 84.0 | 100.0 | 82.1 | 100.0 | 74.0 | 100.0 | 80.0 | 100.0 | 80.1 | 100.0 | 88.8 | 100.0 | 94.1 | 92.2 | 87.7 | 97.4 |
| PGLLM$^O$ (Ours) | PointLLM-7B | GPT-4 | 87.3 | 56.6 | 86.2 | 44.3 | 79.2 | 60.8 | 84.3 | 53.9 | 79.7 | 55.8 | 90.9 | 41.3 | 96.0 | 26.5 | 88.9 | 41.2 |
| PGLLM$^T$ (Ours) | PointLLM-7B | GPT-4 | 89.6 | 53.1 | 87.2 | 43.0 | 80.8 | 60.2 | 85.9 | 52.1 | 81.8 | 52.4 | 93.9 | 26.7 | 97.6 | 9.7 | 91.1 | 29.6 |
| PGLLM$^T$ (Ours) | PointLLM-7B | DeepSeek-V3 | 86.4 | 70.0 | 83.6 | 59.1 | 76.2 | 68.2 | 82.1 | 65.8 | 83.8 | 62.3 | 91.7 | 36.4 | 97.1 | 18.6 | 90.9 | 39.1 |
| PGLLM$^T$ (Ours) | PointLLM-7B | Qwen-Plus | 86.5 | 68.4 | 84.2 | 47.3 | 77.9 | 71.7 | 82.9 | 62.5 | 81.4 | 61.8 | 92.8 | 33.0 | 97.9 | 11.1 | 90.7 | 35.3 |
| MiniGPT-3D(Tang et al., 2024) | MiniGPT-3D | GPT-4 | 86.7 | 100.0 | 84.6 | 100.0 | 79.9 | 100.0 | 83.7 | 100.0 | 62.0 | 100.0 | 87.5 | 100.0 | 93.6 | 100.0 | 81.0 | 100.0 |
| PGLLM$^T$ (Ours) | MiniGPT-3D | GPT-4 | 92.0 | 44.0 | 89.2 | 36.9 | 83.7 | 54.2 | 88.1 | 45.0 | 80.3 | 100.0 | 94.3 | 22.4 | 97.8 | 12.4 | 90.8 | 44.9 |

Table 1: Evaluation of 3D OOD detection on ModelNet40 and ShapeNetCore. **Bold** and underlined numbers denote the best and second-best results, respectively. Each "MNx" or "SNx" denotes the known class split and the rest are unknown.

| Method | one Stage LLM | 2nd Stage LLM | ModelNet40 | | | | | | | | ShapeNetCore | | | | | | | |
|---|---|---|---|---|---|---|---|---|---|---|---|---|---|---|---|---|---|---|
| | | | MN1 | | MN2 | | MN3 | | Average | | SN1 | | SN2 | | SN3 | | Average | |
| | | | AUROC↑ | FPR95↓ | AUROC↑ | FPR95↓ | AUROC↑ | FPR95↓ | AUROC↑ | FPR95↓ | AUROC↑ | FPR95↓ | AUROC↑ | FPR95↓ | AUROC↑ | FPR95↓ | AUROC↑ | FPR95↓ |
| PGLLM$^T$ | PointLLM-7B | Qwen3-VL-8B | 85.8 | 67.4 | 84.4 | 47.5 | 74.2 | 71.3 | 81.5 | 62.0 | 81.2 | 48.0 | 91.9 | 30.3 | 94.4 | 29.7 | 89.2 | 36.0 |
| PGLLM$^T$ | PointLLM-7B | Llama3.1-8B | 75.6 | 86.5 | 72.2 | 82.4 | 60.9 | 91.9 | 69.6 | 86.9 | 56.3 | 89.7 | 81.9 | 68.3 | 92.7 | 44.6 | 80.0 | 67.5 |
| PGLLM$^T$ | PointLLM-7B | GPT-oss-20B | 83.0 | 80.4 | 74.5 | 76.8 | 72.6 | 94.0 | 76.7 | 83.7 | 81.9 | 57.1 | 92.4 | 30.3 | 96.7 | 21.2 | 90.3 | 36.2 |
| PGLLM$^T$ | MiniGPT-3D | Qwen3-VL-8B | 92.4 | 53.0 | 86.4 | 44.5 | 80.0 | 59.2 | 86.3 | 52.2 | 79.1 | 51.0 | 91.4 | 29.2 | 93.8 | 35.5 | 88.1 | 38.6 |
| PGLLM$^T$ | MiniGPT-3D | Llama3.1-8B | 81.8 | 85.9 | 73.0 | 84.0 | 71.9 | 93.1 | 75.6 | 87.7 | 51.7 | 100.0 | 77.2 | 73.0 | 91.6 | 56.9 | 73.5 | 76.6 |
| PGLLM$^T$ | MiniGPT-3D | GPT-oss-20B | 92.0 | 46.9 | 87.9 | 41.2 | 82.6 | 57.9 | 87.5 | 48.7 | 85.9 | 42.6 | 93.2 | 25.0 | 97.0 | 22.1 | 92.0 | 29.9 |

Table 2: Evaluation of 3D OOD detection on ModelNet40 and ShapeNetCore for open-sourced LLM.

# 4 EXPERIMENTS

## 4.1 EXPERIMENTAL SETUP

**Dataset**: We evaluate our method on four well-established 3D point cloud benchmarks. **Model-Net40** (Wu et al., 2015) contains 2,468 test 3D objects across 40 categories. **ShapeNetCore** (Chang et al., 2015) s a canonical subset of the full ShapeNet repository with 5,158 unique test models from 55 object categories. Following PointLLM (Xu et al., 2024), we sample 200 objects from **Objaverse** (Deitke et al., 2023) for testing. **S3DIS** (Armeni et al., 2016) provides semantically segmented 3D point clouds from indoor environments and we follow Chen et al. (2025) to select 8,931 point clouds with rich semantic annotations for evaluation. For OOD dataset partitioning, we follow the 3DOS protocol Alliegro et al. (2022) to divide ShapeNetCore into SN1, SN2, and SN3 subsets. Similarly, we partition ModelNet40 into MN1, MN2, and MN3 subsets. Further dataset details are provided in the supplementary material.

**Implementation Details**: We evaluate PGLLM on two baseline works: PointLLM-7B (Xu et al., 2024) and MiniGPT-3D (Tang et al., 2024) to generate initial captions, and use Point-BERT (Yu et al., 2021b) as the 3D encoder. We evaluate DeepSeek-V3 (Liu et al., 2024), Qwen-Plus (Yang et al., 2024) and GPT-4 Achiam et al. (2023) as the second stage LLMs. To further validate the feasibility and portability of PGLLM, we additionally conduct experiments with open-source LLMs, including Qwen3-VL-8B (Yang et al., 2025a), Llama3.1-8B (Grattafiori et al., 2024), and GPT-oss-20B (Agarwal et al., 2025). These models are integrated into the second-stage reasoning module under the same prompting and inference protocol, enabling a controlled comparison against proprietary counterparts. As for KNN Graph construction, we set the K-value of 3. For score propagation, we set the $\alpha$ to 0.5, and the number of iterations $T$ to 5 in Eq. 2.

**Competing Methods**: We compare against existing LLM-based 3D understanding methods: InstructBLIP (Dai et al., 2023), LLaVA (Liu et al., 2023), 3D-LLM (Hong et al., 2023), Point-Bind LLM (Guo et al., 2023), ShapeLLM (Qi et al., 2024a), PointLLM (Xu et al., 2024), and MiniGPT-3D (Tang et al., 2024). Both PointLLM and MiniGPT-3D employ ChatGPT 4 as second stage LLM for classification, thus facilitating fair comparison. To the best of our knowledge, this is the first work to explore 3D OOD detection within an LLM framework. Therefore, we compare against several VLM-based zero-shot OOD baselines: MCM (Ming et al., 2022), NegLabel (Jiang et al., 2024a), ZLap (Kalantidis et al., 2024), and GSP (Chen et al., 2025). Finally, we evaluate two variants of PGLLM with different support set $\mathcal{D}_u$. When testing data distribution is available, i.e. transductive inference, we use all testing data as the support dataset and refer to the method as PGLLM$^T$. Alternative, we leverage an external dataset, Objaverse, to build the support dataset and refer to the method as PGLLM$^O$. Specifically, we randomly selected 100K samples and their corresponding captions from the 660K training data of Objaverse to build the graph. For both PGLLM$^O$ and PGLLM$^T$, PointLLM-7B is used to generate initial captions. All competing methods use the same 3D encoder as PointLLM (Xue et al., 2024).

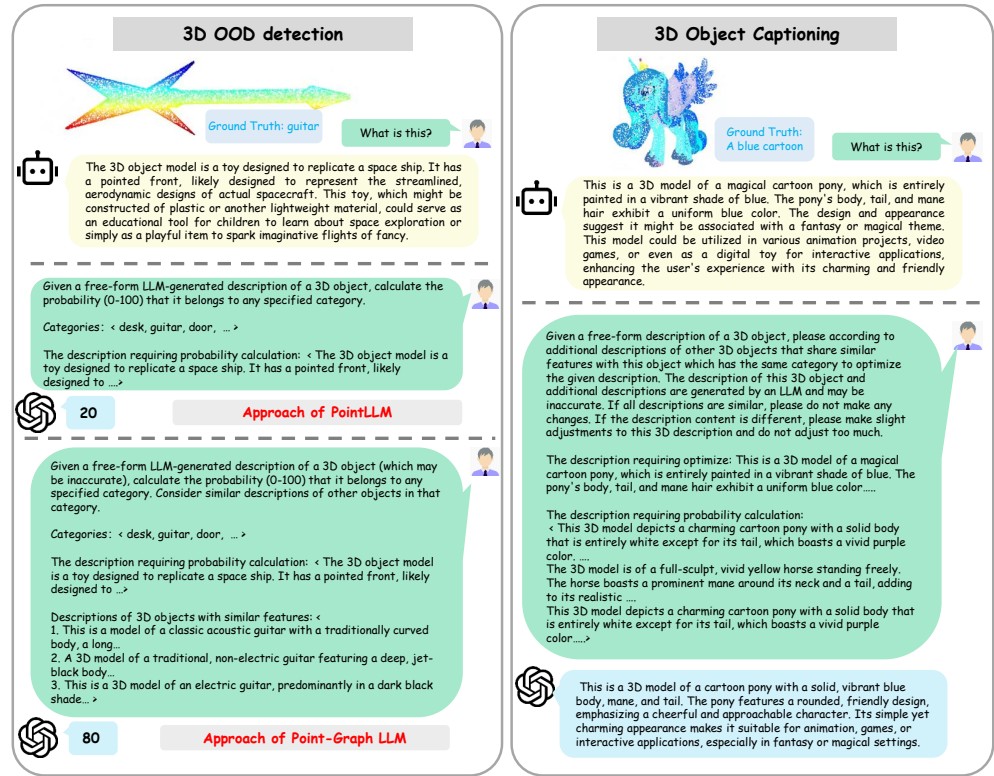

Figure 3: Qualitative examples of prompt questions and PGLLM's predictions.

**Evaluation Metrics** We use classification accuracy (ACC) for 3D recognition, and AUROC and FPR95 for 3D OOD detection, which are standard metrics in OOD evaluation. For 3D object captioning, we assess semantic alignment using Sentence-BERT (Reimers & Gurevych, 2019), Sim-CSE (Gao et al., 2021), and GPT-4 as evaluators.

## 4.2 EXPERIMENTAL RESULTS

**3D OOD Detection:** Tab. 1 summarizes the results of our comprehensive experiments, highlighting the following key observations. **(i)** Our PGLLM[T] framework, when integrated with GPT-4, establishes new state-of-the-art results on the ModelNet40 benchmark. It achieves an outstanding 85.9% AUROC on average, outperforming the previous best method (MCM) by 4.9%, while also reducing the critical FPR95 metric to 52.1%. We notice that both PointLLM-7B and ours methods use ChatGPT 4 as the second stage LLM. **(ii)** On ShapeNetCore, PGLLM[T] with GPT-4 demonstrates breakthrough performance, attaining 97.6% AUROC on SN3 and a remarkably low FPR95 of 9.7%. This reflects an 7.1% AUROC improvement over feature-based methods such as GSP. Overall, our framework achieves an average AUROC of 91.1% and an average FPR95 of 29.6%, setting new benchmarks across key metrics. These consistent gains across two datasets affirm the effectiveness of our graph-based mechanism. **(iii)** Because GPT-4 tends to assign either 0 or 100 when scoring test samples, the baseline performs very poorly on the FPR95 metric, with values almost always equal to 100.0. By introducing score propagation, our method effectively alleviates this issue and yields much smoother, more evenly distributed scores across all samples. **(iv)** Using GPT-4 as the second stage LLM yields notable improvements in AUROC with +3.8% on ModelNet40 and +0.2% on ShapeNetCore compared to DeepSeek-V3. The performance gap between GPT-4, Qwen-Plus and DeepSeek-V3 variants highlights the ability of our framework to harness stronger LLMs for enhanced 3D understanding. **(v)** With both testing data (transductive setting) and external dataset as supporting dataset, PGLLM[T/O] outperforms the baseline (PointLLM-7B). This suggest the robustness of the graph-based method.

**Evaluated on Open-sourced LLM:** To further validate the practicality of our PGLLM beyond closed-source APIs, we additionally evaluate our framework with representative open-sourced

| Method | 2nd Stage LLM | 3D Recognition | | | 3D Captioning | | |
|---|---|---|---|---|---|---|---|
| | | (I) ACC | (C) ACC | Average | GPT-4 | S-BERT | SimCSE |
| 3D-LLM(Hong et al., 2023) | - | - | - | - | 33.4 | 44.5 | 43.7 |
| Point-Blind(Guo et al., 2023) | - | 51.9 | 39.7 | 45.8 | - | - | - |
| ShapeLLM-7B(Qi et al., 2024a) | - | - | - | - | 46.9 | 48.2 | 49.2 |
| ShapeLLM-13B(Qi et al., 2024a) | - | - | - | - | 49.0 | 48.5 | 50.0 |
| InstructBLIP-7B(Dai et al., 2023) | GPT-4 | 19.5 | 31.5 | 25.5 | 45.3 | 47.4 | 48.5 |
| InstructBLIP-13B(Dai et al., 2023) | GPT-4 | 26.0 | 31.4 | 28.7 | 45.0 | 45.9 | 48.9 |
| LLaVA-7B(Liu et al., 2023) | GPT-4 | 39.7 | 39.7 | 39.7 | 46.7 | 45.6 | 47.1 |
| LLaVA-13B(Liu et al., 2023) | GPT-4 | 37.1 | 36.1 | 36.6 | 38.3 | 46.4 | 45.9 |
| MiniGPT-3D(Tang et al., 2024) | GPT-4 | 61.8 | 60.0 | 60.9 | 57.1 | 49.5 | 51.4 |
| PointLLM-7B(Xu et al., 2024) | GPT-4 | 53.4 | 51.8 | 52.6 | 44.9 | 47.5 | 48.6 |
| PointLLM-13B(Xu et al., 2024) | GPT-4 | 53.0 | 52.6 | 52.8 | 48.2 | 47.9 | 49.1 |
| PGLLM$^O$ (Ours) | GPT-4 | 53.1 | 53.0 | 53.1 | 49.1 | 48.4 | 48.9 |
| PGLLM$^T$ (Ours) | GPT-4 | 63.1 | 61.8 | 62.5 | 50.5 | 48.9 | 49.4 |
| PGLLM$^T$ (Ours) | DeepSeek-V3 | 62.6 | 62.0 | 62.3 | - | - | - |
| PGLLM$^T$ (Ours) | Qwen-Plus | 43.1 | 41.7 | 42.4 | - | - | - |

Table 3: Comparison of results on 3D recognition (ModelNet40) and 3D captioning (Objaverse). Recognition performance is evaluated using two prompt types: an Instruction-type (I) prompt ("*What is this?*") and a Completion-type (C) prompt ("*This is an object of*").

LLMs as the second-stage inference module: Qwen3-VL-8B, Llama3.1-8B, and GPT-oss-20B. As shown in Tab. 2, PGLLM remains consistently effective across different open-source choices, demonstrating that our improvements primarily come from the proposed graph-based in-context guidance and score refinement, rather than relying on any specific proprietary model. Importantly, the best open-source configurations can achieve performance that is close to, or even surpasses the closed-source inference results in certain splits, using MiniGPT-3D features with Qwen3-VL-8B or GPT-oss-20B yields strong average AUROC on both ModelNet40 and ShapeNetCore, highlighting the competitiveness of our method under open-source inference. We also observe that Llama3.1-8B is generally weaker, which we attribute to its chat-oriented nature: it is optimized for conversational instruction-following and tends to be less reliable for the numeric, reasoning-intensive probability scoring required by our OOD setup. Overall, these results confirm that PGLLM is robust and transferable, enabling strong 3D OOD detection with open-sourced LLMs at a level comparable to closed-source inference.

**3D Recognition:** The results in Tab. 3 reveal several key insights. **i)** Our PGLLM$^T$ framework achieves an average accuracy of 62.3% with DeepSeek-V3 and 62.5% with GPT-4, outperforming all existing state-of-the-art methods. In particular, it surpasses the strongest baseline, MiniGPT-3D, by +1.6% for average. **ii)** The framework shows strong robustness across different prompt types. Specifically, it outperforms MiniGPT-3D by 1.3% with the instruction prompt "What is this?" and by 1.8% with the completion prompt "This is an object of". This dual-prompt advantage highlights the adaptability of our architecture to diverse query formats. **iii)** Although all competing methods rely on GPT-4 for evaluation, our framework achieves 62.3% average accu-

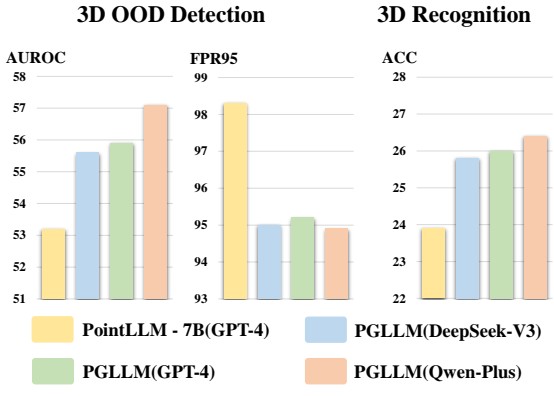

Figure 4: Results on real-world benchmark S3DIS. We report 3D OOD detection and 3D recognition tasks.

racy even with DeepSeek-V3 (much lower per token cost than GPT-4) as second stage LLM, surpassing all GPT-4-based baselines. This suggests lower cost LLMs may achieve comparable performance. However, Qwen-Plus demonstrates notably lower performance on the 3D Recognition task, primarily due to its limited ability to generate long numerical sequences, which adversely affects its overall results.

**3D Object Captioning:** The results in Tab. 3 show that our method performs competitively on the 3D object captioning task, consistently outperforming the PointLLM-7B baseline across all three

| In-context Guidance | Score Propagation | ModelNet40 | | | ShapeNetCore | | |
|---|---|---|---|---|---|---|---|
| | | ACC↑ | AUROC↑ | FPR95↓ | ACC↑ | AUROC↑ | FPR95↓ |
| - | - | 52.5 | 80.4 | 100.0 | 55.5 | 88.2 | 54.9 |
| ● | - | 59.7 | 83.3 | 100.0 | 60.7 | 89.2 | 47.2 |
| ✓ | - | 60.2 | 83.1 | 100.0 | 61.0 | 89.5 | 46.0 |
| - | ✓ | 56.7 | 83.5 | 62.0 | 59.3 | 89.8 | 44.7 |
| ✓ | ✓ | 63.1 | 85.9 | 52.1 | 62.4 | 91.1 | 29.6 |

Table 4: Ablation study on two datasets. ACC refers to the results of 3D recognition experiments, while AUROC and FPR95 correspond to the OOD detection experiments. Both AUROC and FPR95 represent averages across all subsets of the ModelNet40 and ShapeNetCore datasets. The ● denotes in-context guidance derived through direct nearest-sample retrieval without graph.

evaluation metrics. Since the evaluation relies on subjective judgments from an LLM, PGLLM uses GPT-4 exclusively for assessment to ensure a consistent evaluation standard. While our approach does not yet reach state-of-the-art performance, this gap is likely attributable to the limited size of the test dataset. In low-data regimes, constructing optimal graph structures is more challenging, which can constrain performance. Nevertheless, the observed improvements highlight the promise of our graph-based in-context guidance strategy for 3D captioning tasks. This suggests careful selection of second stage LLM is still necessary.

**Experiments on Real-world Benchmark:** Most existing 3D LLMs Xu et al. (2024); Guo et al. (2023); Tang et al. (2024) have been evaluated primarily on synthetic CAD datasets, leaving a validation gap in real-world scenarios. To address this, we conduct experiments on the real-world S3DIS dataset. As shown in Fig. 4, we evaluate both PointLLM and PGLLM on 3D out-of-distribution (OOD) detection and 3D recognition tasks. The results demonstrate that our method consistently outperforms baseline approaches, confirming the effectiveness of our graph-based strategy in practical settings. Notably, PGLLM remains compatible with lower-cost second-stage LLMs, achieving performance that is comparable, or even superior to higher-cost alternatives.

## 4.3 ABLATION STUDY

We conduct an ablation study with results summarized in Tab. 4. All experiments use ChatGPT4 as the second-stage LLM and are conducted on the ModelNet40 and ShapeNet-Core datasets for both 3D recognition and 3D OOD detection. We begin with a baseline that feeds the generated 3D caption of each query directly into the LLM without additional context. We also evaluate a variant where the query retrieves its nearest neighbors via KNN ($K = 3$) to form the in-context prompt, by-passing graph construction.

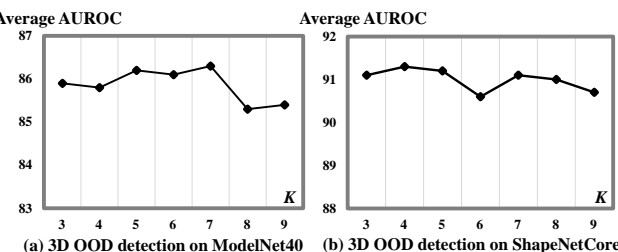

(a) 3D OOD detection on ModelNet40    (b) 3D OOD detection on ShapeNetCore

Figure 5: Different number of K-values for 3D OOD detection on two datasets.

This variant yields results comparable to the graph-based method, likely because both retrieve highly similar neighbors due to the small $K$. However, omitting the graph structure prevents the use of score propagation for further refinement. Introducing in-context guidance alone yields significant performance gains over the baseline, demonstrating its effectiveness in injecting relevant contextual cues into the LLM. In contrast, score propagation alone provides only modest improvements. Notably, combining in-context guidance with score propagation leads to substantial performance boosts across both tasks, underscoring their complementary and synergistic effects.

## 4.4 FURTHER ANALYSIS

**Qualitative Analysis**: Fig. 3 showcases qualitative examples of PGLLM on 3D OOD detection and 3D object captioning tasks. For 3D OOD detection, we observe that PointLLM's reliance on single-caption inputs leads to fragile predictions when faced with ambiguous or hard-to-interpret

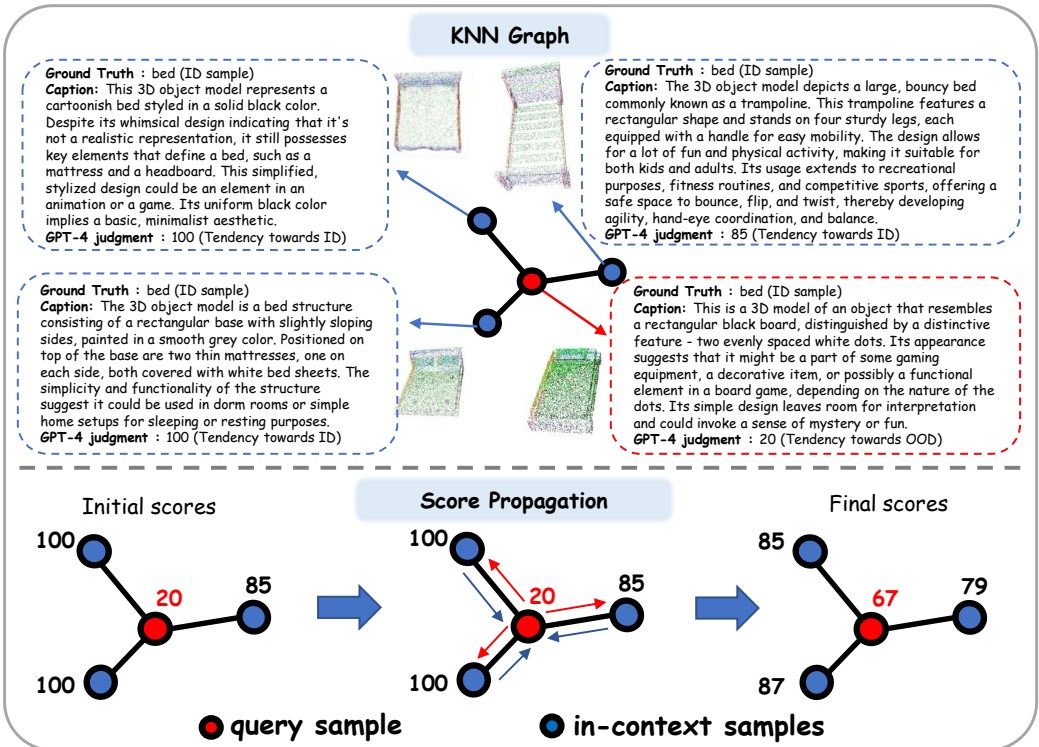

Figure 6: Qualitative example demonstrating the effectiveness of score propagation.

point clouds. Our in-context guidance mitigates this by providing captions from structurally similar samples, allowing the LLM to leverage contextual cues for more robust classification. For 3D object captioning, PointLLM generates captions exhibiting inaccuracies or redundancies. By incorporating in-context guidance, our method enables the LLM to refine these outputs based on contextual similarity, resulting in more accurate and semantically rich descriptions. Fig. 6 qualitatively illustrates the process of score propagation. It can be observed that when the 3D LLM generates a low-quality caption for a given sample, it directly leads to a poor score in the second-stage evaluation. However, by employing the score propagation method, this can be effectively corrected by leveraging the accurate scores from neighboring samples of the query sample.

**Impact of In-Context Example Quantity:** To examine the effect of in-context guidance, we vary $K$, the number of retrieved captions provided to the LLM, and evaluate 3D OOD detection on ModelNet40 and ShapeNetCore in Fig. 5. Performance improves as $K$ increases, reaching a dataset-specific peak at $K = 7$ for ModelNet40 and $K = 4$ for ShapeNetCore. Beyond these values, accuracy declines, likely due to the inclusion of irrelevant or misleading examples that confuse the LLM. These results underscore the importance of selecting an appropriate number of in-context examples for optimal performance.

## 5 CONCLUSION

In this work, we introduce PGLLM, a novel framework for 3D point cloud understanding. Our approach constructs a graph to retrieve structurally similar 3D captions for each point cloud, using them as in-context examples to guide the LLM toward more informed reasoning. In parallel, a score-based refinement mechanism leverages the intrinsic structure of the test data manifold to enhance prediction accuracy. PGLLM achieves competitive results across a range of downstream tasks and sets new state-of-the-art performance in both 3D out-of-distribution detection and 3D recognition. Extensive experiments on diverse 3D point cloud datasets demonstrate the robustness and generalizability of our method. We believe this work offers a promising direction for advancing 3D point cloud understanding and integrating LLMs into spatial perception tasks.

## Acknowledgments

This research is supported by the National Natural Science Foundation of China (No. U2468207), Sichuan Science and Technology Program (No. 2024NSFTD0036) and Agency for Science, Technology and Research (A*STAR) under its MTC Programmatic Funds (Grant No. M23L7b0021).

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

# A APPENDIX

## A.1 OVERVIEW OF THE TASKS

We illustrate the three tasks addressed in this work in Fig. 7. Our framework integrates 3D point clouds with large language models by first constructing prompts using PointLLM and a point cloud feature graph (Step 1), then leveraging LLMs to perform downstream tasks (Step 2). PointLLM generates free-form descriptions of 3D objects (yellow boxes), which are refined and utilized by the LLM for tasks including 3D out-of-distribution (OOD) detection, 3D recognition, and 3D object captioning.

## A.2 DATASET PARTITION FOR 3D OOD DETECTION

We follow the benchmark 3DOS, GSP to construct ShapeNetCore and S3DIS datasets for OOD detection. For the ModelNet40 dataset, we follow the partitioning practice established in the 3DOS work, dividing it into three subsets based on ascending index order. The specific category partitioning details are presented in Tab 7 5 6. Qualitative visualizations of some dataset samples are presented in Fig. 8 10.

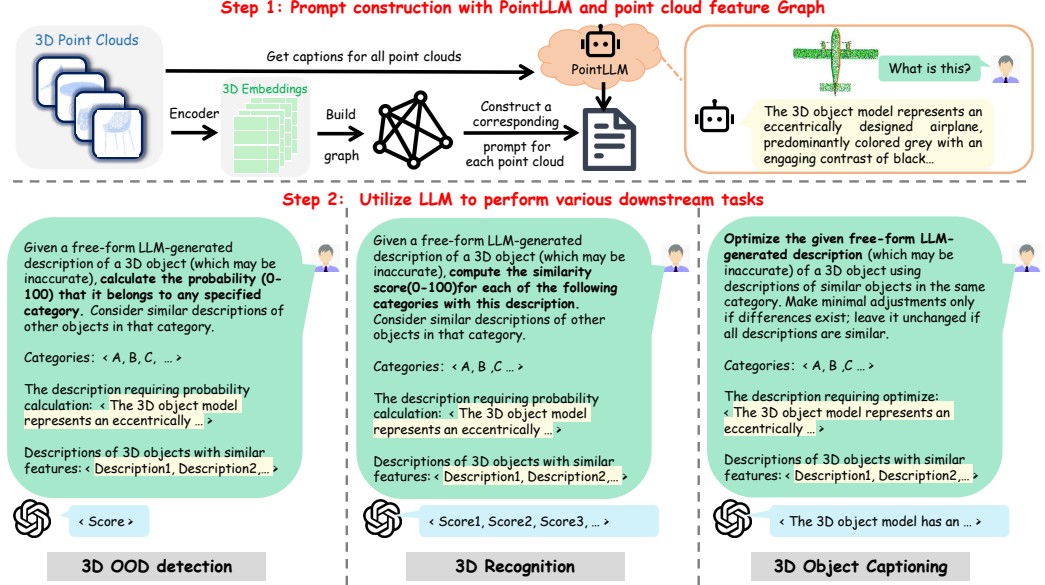

Figure 7: Demonstrations of PGLLM. We propose PGLLM, an efficient and potent framework that integrates 3D-LLMs with Large Language Models, where the text on a light yellow background indicates content generated by PointLLM. Furthermore, we demonstrate its operational mechanisms across 3D recognition, 3D OOD detection and 3D object captioning tasks.

| SN1 | mug, lamp, bed, washer, loudspeaker, telephone, dishwasher, camera, birdhouse, jar, bowl, bookshelf, stove, bench, display, keyboard, clock, piano |
|---|---|
| SN2 | earphone, knife, chair, pillow, table, laptop, mailbox, basket, file cabinet, cabinet, sofa, printer, flowerpot, microphone, tower, bathtub, bag, trash bin |
| SN3 | can, microwave, skateboard, faucet, train, guitar, pistol, helmet, watercraft, airplane, bottle, cap, rocket, rifle, remote, car, bus, motorbike |

Table 5: For each distinct out-of-distribution (OOD) subset partition on the ShapeNetCore, the categories residing within a given subset are designated as in-distribution (ID), whereas categories from all other subsets are considered entirely OOD.

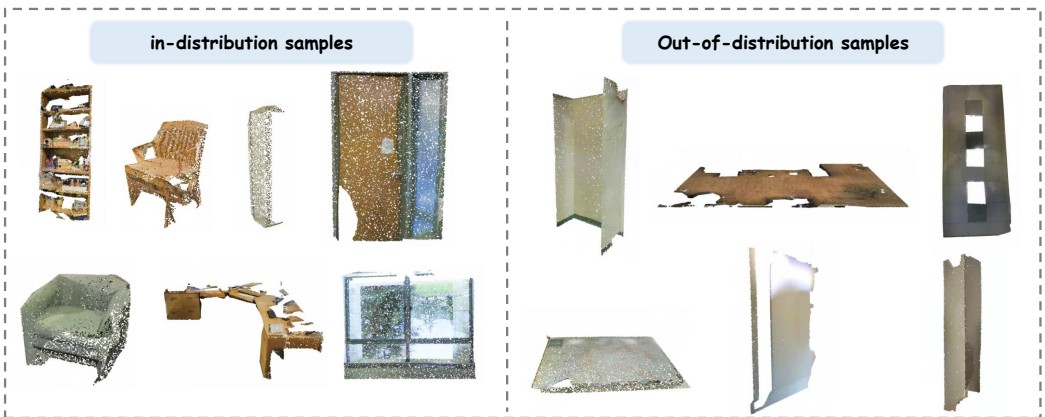

Figure 8: Visualization of the real-world benchmark S3DIS and the categories partition for OOD detection.

| ID | window, door, table, chair, sofa, bookcase, clutter |
|---|---|
| OOD | beam, board, ceiling, column, floor, wall |

Table 6: S3DIS dataset partitioning for OOD detection: foreground objects as ID and background objects as OOD.

### A.3  PROMPTS AND QUALITATIVE ANALYSIS

We provide the prompts for 3D recognition, 3D OOD detection and 3D object captioning tasks in Tab. 12. Furthermore, we provide supplementary visual analyses for 3D recognition and 3D OOD detection tasks in Fig. 11, 12, 13and 14, provided with comprehensive prompts to the LLM. As evidenced in Fig. 11, the query sample contains an erroneous caption, yet the collective accuracy of "vase" annotations within the in-context guidance enables the LLM to yield the correct prediction.Fig. 13 illustrates a scenario where both the query sample and several in-context guidance captions contain misleading annotations. This induces the LLM to output a neutral score. Nevertheless, this outcome represents some improvement over the erroneous predictions generated without in-context guidance. Despite the fact that the query sample's caption lacks direct information about the categories, as illustrated in Fig. 14, the captions of the in-context samples prove crucial for enabling the LLM to achieve a more accurate classification. However, as shown in Fig. 12, the LLM's output is inclined to classify the object as a "lamp," which contradicts the ground-truth label of "faucet." This discrepancy is likely attributable to the presence of the word "lights" in the query sample's caption. Despite the in-context samples providing captions containing "faucet," the LLM ultimately assigned a full score of 100 to the "lamp" category, compared to only 60 for "faucet." In Fig. 15, we present examples where in-context guidance exerts a negative impact. It can be observed that when the caption generated for the query sample is relatively accurate, but the captions of the in-context samples contain certain inaccuracies, the second-stage scoring becomes biased.

| MN1 | airplane, bathtub, bed, bench, bookshelf, bottle, bowl, car, chair, cone, cup, curtain, desk |
|---|---|
| MN2 | door, dresser, flower pot, glass box, guitar, keyboard, lamp, laptop, mantel, monitor, night stand, person, piano |
| MN3 | plant, radio, range hood, sink, sofa, stairs, stool, table, tent, toilet, tv stand, vase, wardrobe, xbox |

Table 7: For each distinct out-of-distribution (OOD) subset partition on the ModelNet40, the categories residing within a given subset are designated as in-distribution (ID), whereas categories from all other subsets are considered entirely OOD.

| 50% samples | | 60% samples | | 70% samples | | 80% samples | | 90% samples | | 100% samples | |
|---|---|---|---|---|---|---|---|---|---|---|---|
| AUROC ↑ | FPR95 ↓ | AUROC ↑ | FPR95 ↓ | AUROC ↑ | FPR95 ↓ | AUROC ↑ | FPR95 ↓ | AUROC ↑ | FPR95 ↓ | AUROC ↑ | FPR95 ↓ |
| 86.5 | 59.3 | 87.4 | 57.4 | 87.2 | 57.5 | 87.0 | 59.6 | 86.5 | 61.9 | 86.8 | 58.3 |

Table 8: The Dynamic Graph Expansion experiments conducted on MN1.

| Cost | GPT-4 (USD $) | DeepSeek V3 (CNY ¥ / USD $) | Qwen-Plus (CNY ¥ / USD $) |
|---|---|---|---|
| Total (2,468 samples) | $28.00 | ¥4 / $0.56 | ¥7 / $0.98 |
| Per 1,000 samples | $11.30 | ¥1.6 / $0.23 | ¥2.8 / $0.39 |
| AUROC (%) | 85.9 | 82.1 | 82.9 |

Table 9: Cost comparison of different second-stage LLMs on the OOD detection task. Exchange rate: 1 USD = 7.11 CNY.

## A.4 DYNAMIC GRAPH EXPANSION

To mimic practical deployment scenarios where the full test set is not always accessible at once and the available evaluation data may be dynamically reduced or updated over time, we design an experiment to assess the dynamic graph expansion mechanism under incremental query arrival. Specifically, we randomly split the ModelNet40 testing set into two equal halves (50% each). The initial graph is constructed using first half of the data. Then, from the remaining 50%, we randomly select 10% of the samples as test instances and incrementally add them to the graph for evaluation. Notably, adding new samples does not require reconstructing the entire graph from scratch; instead, we follow the method proposed in Li et al. (2025) to dynamically update the graph structure for the newly added samples. This incremental addition process is repeated five times, and the results after each addition are reported in Tab. 8. These results correspond to the test performance evaluated on the entire accumulated samples after each addition of new samples. Finally, when the sample count reaches 100%, all test samples from the ModelNet40 dataset have been incorporated and evaluated. At this point, AUROC and FPR95 reach 86.8 and 58.3, respectively. In comparison, the results reported in the submission—obtained by performing inference directly on the complete graph constructed from the full dataset—are 89.6 and 53.1 for AUROC and FPR95, respectively. Although there is a moderate performance degradation, our approach still exhibits strong robustness and generalization capability. This further demonstrates that the proposed method does not strictly rely on access to the entire global dataset to achieve effective performance.

## A.5 IMPACT OF K-VALUES

To further explore the influence of K-values, we test different number of $K$ on 3D recognition task. The results are shown in Fig. 9, it can be observed that the optimal performance is achieved at K=5 for ModelNet40 and K=6 for ShapeNet, respectively. Furthermore, performance exhibits a declining trend with increasing K-values, a finding empirically consistent with the conclusions presented in our prior submission.

## A.6 FINANCIAL COST

We evaluated three widely used LLMs as the second-stage model in our OOD inference pipeline and computed their corresponding API costs. Tab. 9 reports the total cost of processing the entire ModelNet40 test set (2,468 samples). GPT-4 incurs a cost of approximately USD $28, whereas DeepSeek-V3 and Qwen-3 Plus cost CNY ¥4 (USD $0.56) and CNY ¥7 (USD $0.98), respectively. All three models are called using API.

This reveals a 30×–50× cost gap between GPT-4 and the two models. Despite this large difference in cost, DeepSeek-V3 and Qwen-3 Plus achieve comparable AUROC performance (82.1–82.9% vs. 85.9% for GPT-4). Given this favorable cost-performance trade-off, practitioners may prefer adopting DeepSeek-V3 or Qwen-3 Plus as the second-stage LLM, especially in large-scale or cost-sensitive deployments.

| | Cost for Existing Approach (PointLLM) | | Additional Cost for PGLLM (Our Method) | | | |
|---|---|---|---|---|---|---|
| | Caption Generation | 2nd stage LLM | Graph Construction | Score Propagation (OOD) | Score Propagation (Recog) | Retrieval for in-context guidance |
| **FLOPs** | 1700G | unknown | 9.6G | 0.12G | 2.4G | 27.8k |
| **Memory** | 14GB | unknown | 46.8MB | 9KB | 0.19MB | 14KB |
| **Param** | 7B | unknown | 0 | 0 | 0 | 0 |
| **Time consumption** | 107mins | 80mins | 126ms | 101ms | 106ms | < 1ms |

Table 10: Computation cost of individual components on ModelNet40.

| | Input data (100K) | embeddings | adjacency | 3D caption |
|---|---|---|---|---|
| **Storage** | 19.2G (in Hard Drive) | 1.24G (in GPU VRAM) | 8M (in GPU) | 87M (in GPU) |

Table 11: Storage cost for 100K samples.

**Inference Time:** Tab. 10 summarizes the computation cost of each component on ModelNet40. Overall, PGLLM introduces negligible extra overhead compared to the LLM-heavy baseline. The added graph construction and score propagation are lightweight, it cost 9.6G FLOPs and 46.8MB memory for graph construction and 0.12–2.4G FLOPs with only KB-level memory for propagation, which taking only 101–126ms in total. In contrast, the existing pipeline is dominated by caption generation and second-stage LLM inference (107mins and 80mins, respectively). This shows PGLLM is computationally efficient and practical in real deployments.

## A.7 DEVICE MEMORY FOR GRAPH STORAGE BUDGETS

In Tab. 11, we summarize the storage requirements of each system component. Notably, 100K samples require only about 19 GB of hard drive storage, while their corresponding embeddings occupy just 1.2 GB when loaded into GPU VRAM for computation. These explicit storage budgets demonstrate that even at a 100K-support scale, both the graph and caption caches fit comfortably within a single GPU. The overall memory footprint is therefore still dominated by the backbone 3D–LLM parameters and activations, rather than by our graph-based test-time scaling modules.

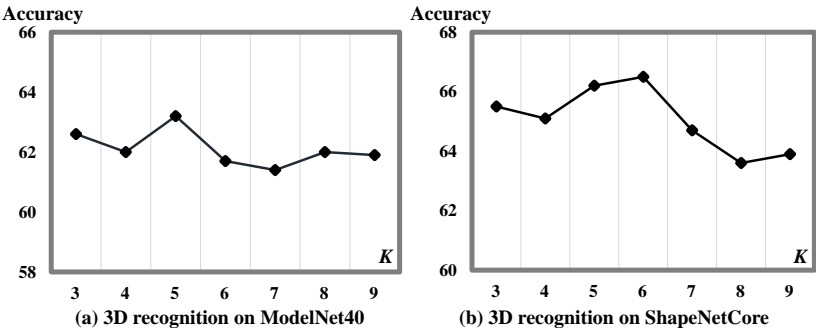

(a) 3D recognition on ModelNet40          (b) 3D recognition on ShapeNetCore

Figure 9: Different number of K-values for 3D recognition on two datasets.

| Tasks | Prompts |
|---|---|
| 3D Recognition | Given a free-form description of a 3D object, the content described here belongs to one of the following 54 categories. Use this description to compute a similarity score (0-100) for each of the following 54 categories. The description of this 3D object is generated by an LLM and may be inaccurate. In addition, I will provide you with descriptions of other 3D objects that share similar features with this object of this category. 0=no relation, 100=perfect match. 

 categories: mug, lamp, bed, washer, loudspeaker, telephone, dishwasher, camera, birdhouse, jar, bowl, bookshelf, stove, bench, display, keyboard, clock, piano, earphone, knife, chair, pillow, table, laptop, mailbox, basket, file cabinet, cabinet, sofa, printer, flowerpot, microphone, tower, bathtub, bag, trash bin, can, microwave, skateboard, faucet, train, guitar, pistol, helmet, watercraft, airplane, bottle, cap, rocket, rifle, remote, car, bus, motorbike. 

 3D object description: ... (# generated by PointLLM) 

 Descriptions of 3D objects with similar features: (# generated by PointLLM and seleted by mainfold learning) 
 1. ... 
 2. ... 
 3. ... 

 Please output the 54 corresponding similarity scores in the order of the above-mentioned categories, without any additional explanation. |
| 3D OOD detection | Given a free-form description of a 3D object, please calculate the probability (0-100) that the content described in the following text pertains to any of the following categories. The description of this 3D object is generated by an LLM and may be inaccurate. In addition, I will provide you with descriptions of other 3D objects that share similar features with this object of this category. You need to take these similar 3D model descriptions into account as well. 0=no relation, 100=perfect match. 

 categories: plant, radio, range hood, sink, sofa, stairs, stool, table, tent, toilet, tv stand, vase, wardrobe, xbox. 

 3D object description: ... (# generated by PointLLM) 

 Descriptions of 3D objects with similar features: (# generated by PointLLM and seleted by mainfold learning) 
 1. ... 
 2. ... 
 3. ... 

 Output only a numerical score. Do not provide additional explanations. |
| 3D Object Captioning | Given a free-form description of a 3D object, please according to additional descriptions of other 3D objects that share similar features with this object which has the same category to optimize the given description. The description of this 3D object and additional descriptions are generated by an LLM and may be inaccurate. If all descriptions are similar, please do not make any changes. If the description content is different, please make slight adjustments to this 3D description and do not adjust too much. 

 The description requiring optimize: ... (# generated by PointLLM) 

 Descriptions of 3D objects with similar features: (# generated by PointLLM and seleted by mainfold learning) 
 1. ... 
 2. ... 
 3. ... 

 Output only a 3D description. And don't describe too much. |

Table 12: A list of prompts for 3D Recognition, 3D OOD Detection, and 3D Object Captioning tasks to LLM.

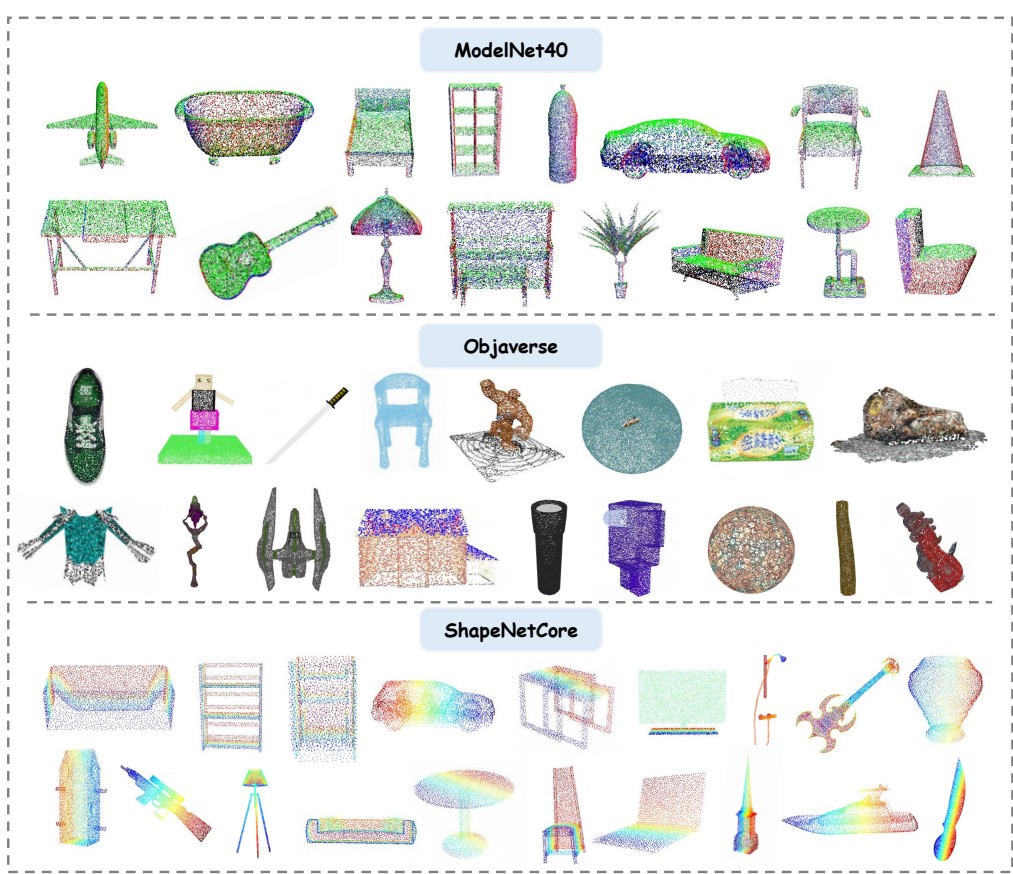

Figure 10: Visualization of the ModelNet40, Objaverse, and ShapeNetCore.

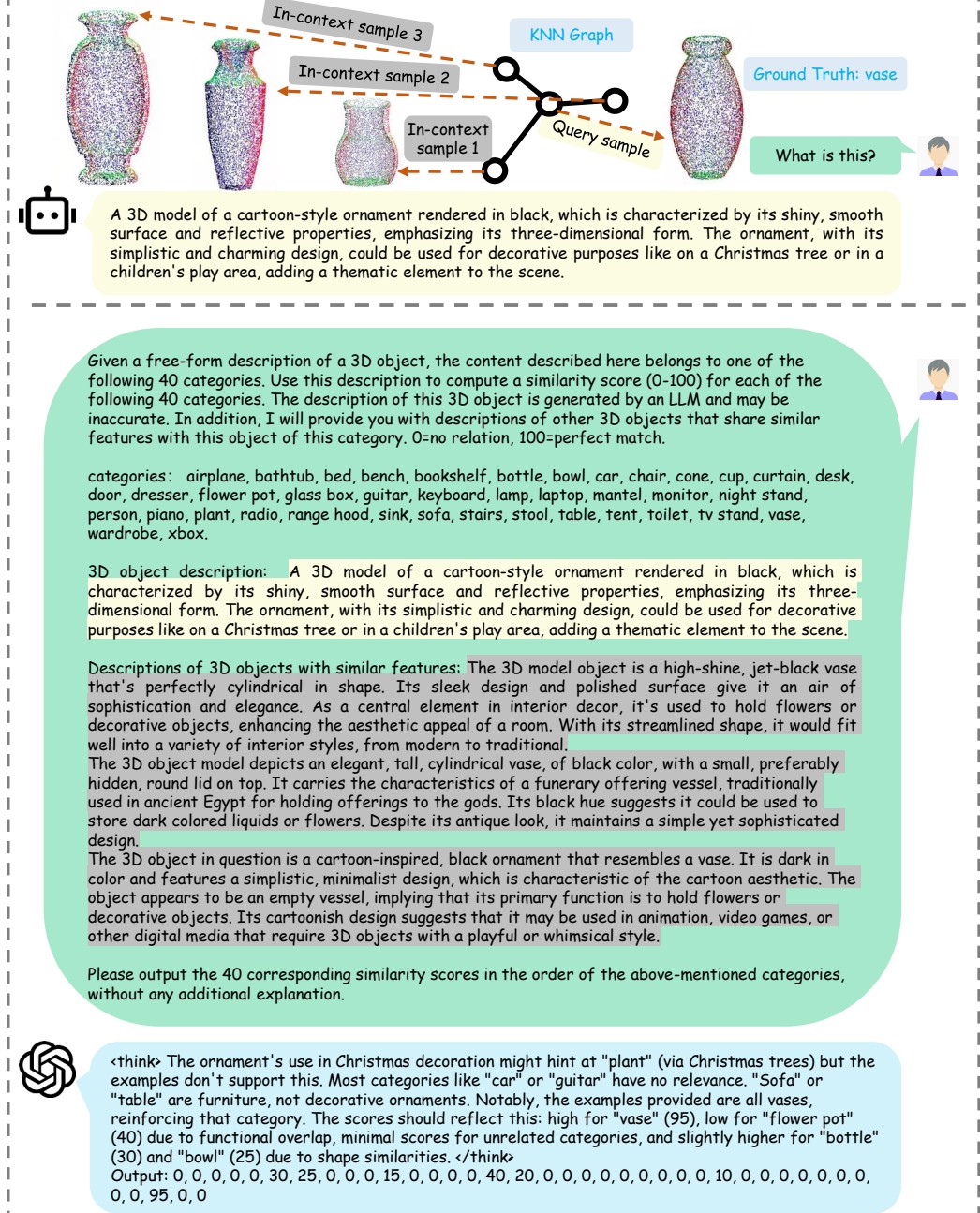

Figure 11: Qualitative example for 3D recognition task on ModelNet40.

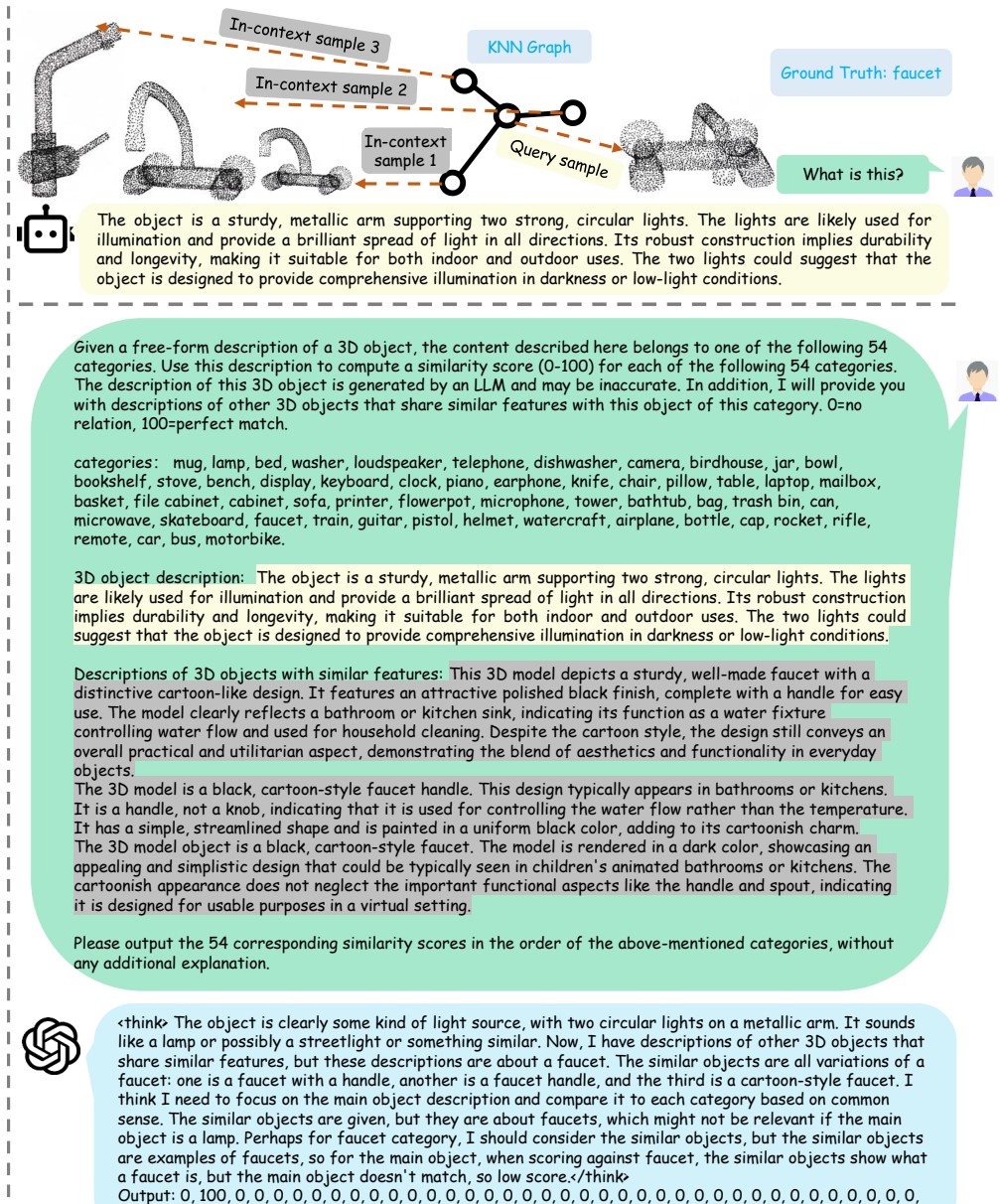

Figure 12: Qualitative example for 3D recognition task on ShapeNetCore.

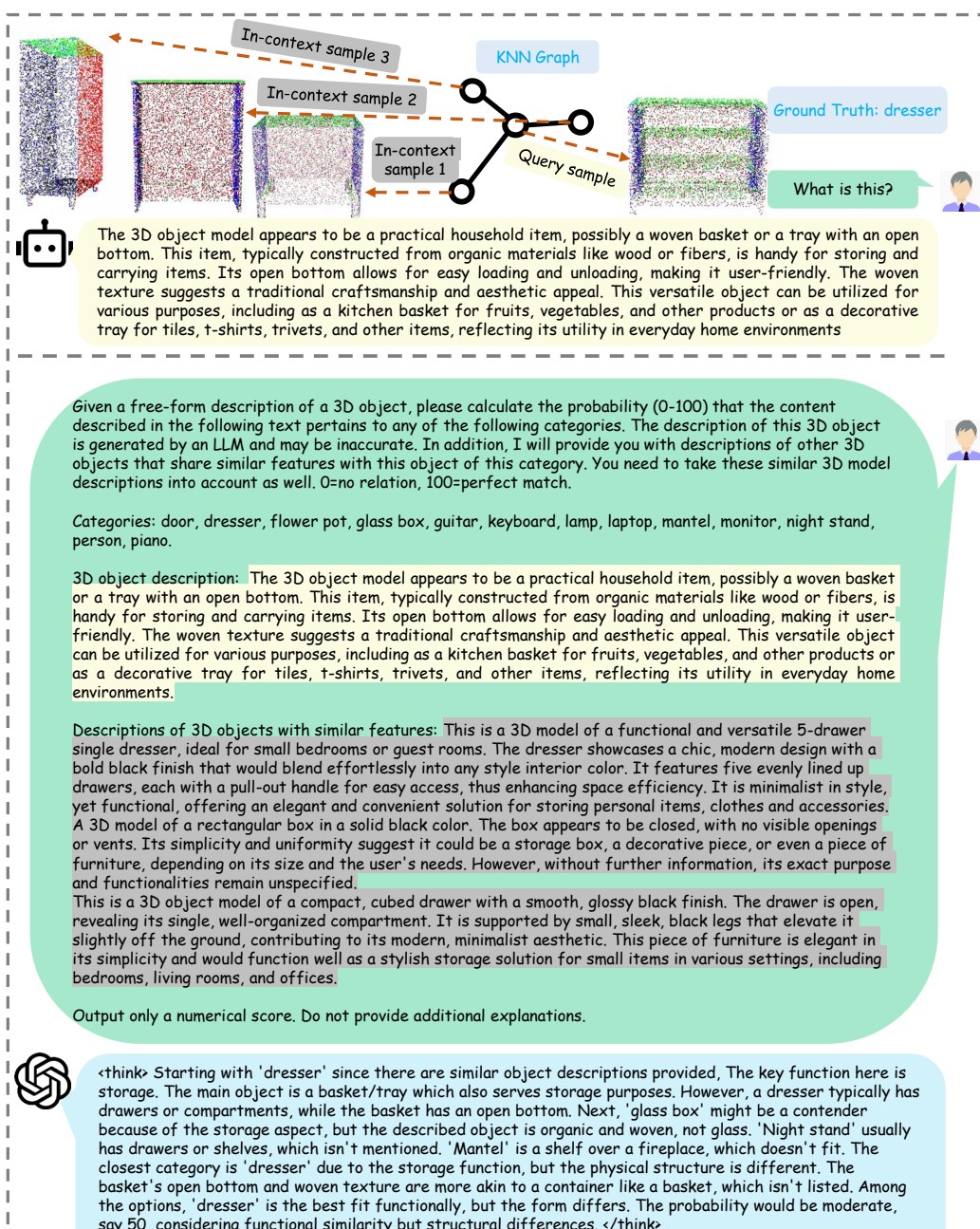

Figure 13: Qualitative example for 3D OOD detection task on ModelNet40.

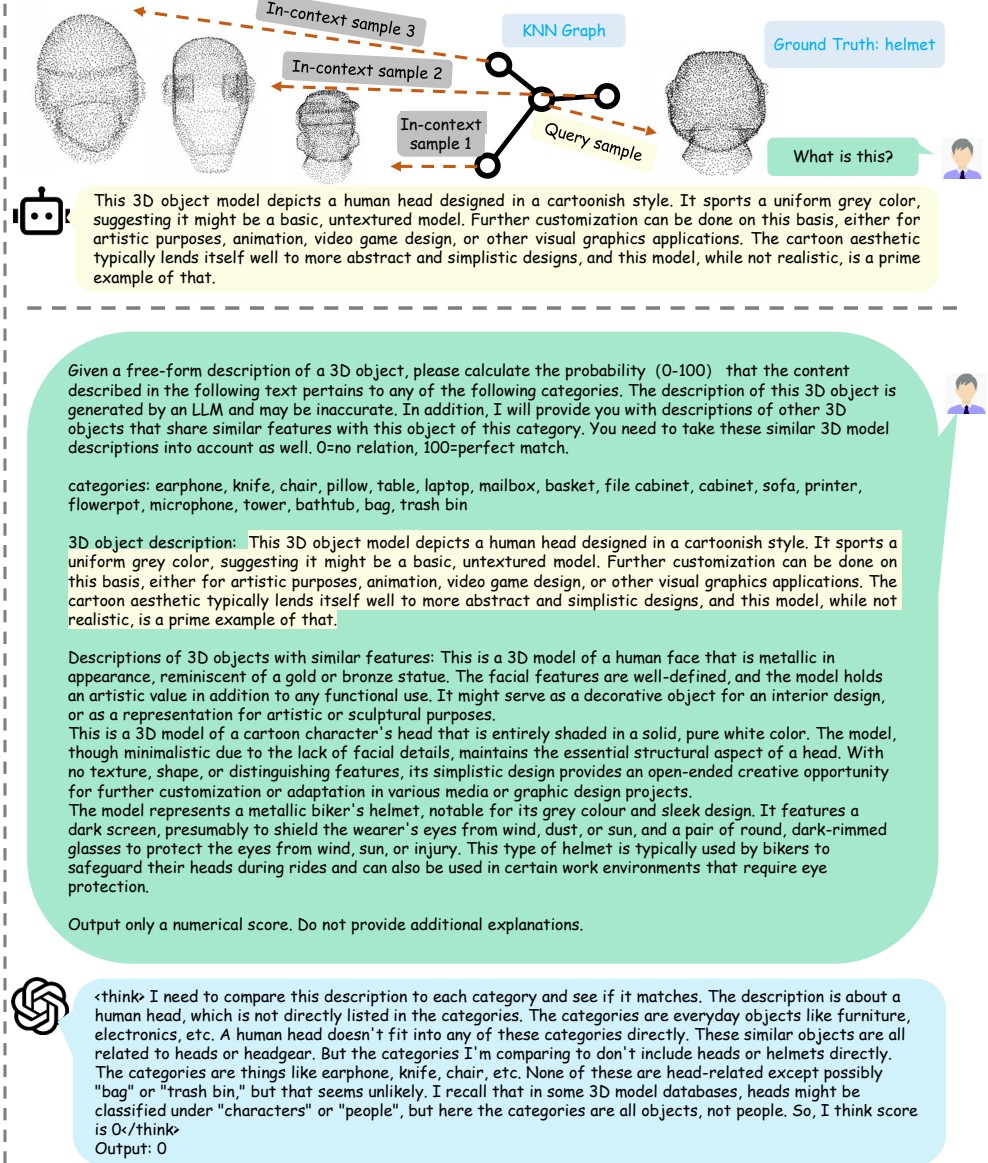

Figure 14: Qualitative example for 3D OOD detection task on ShapeNetCore.

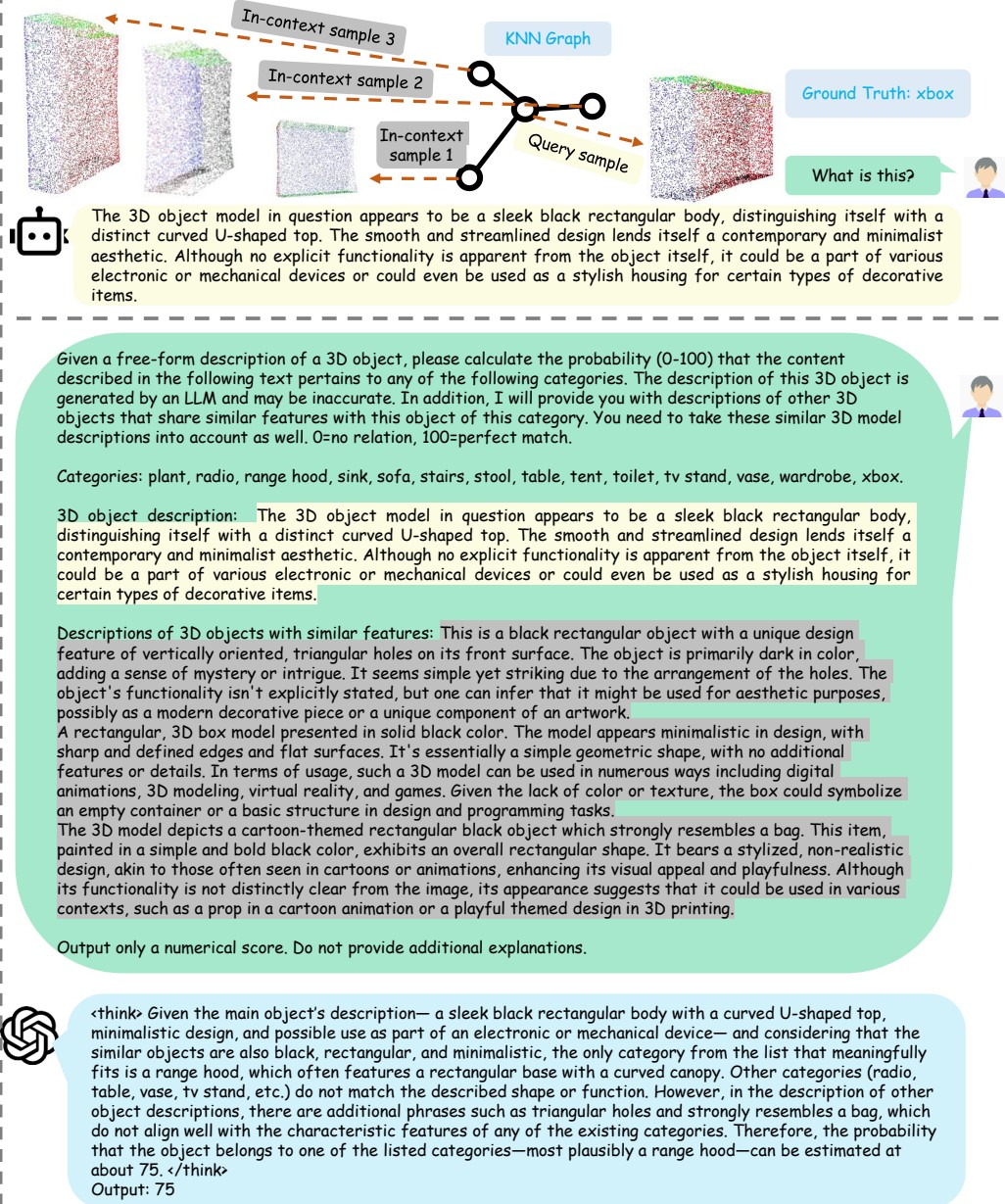

Figure 15: Bad example for 3D OOD detection task on ModelNet40.

