# OpenReview forum: "Test-Time Optimization of 3D Point Cloud LLM via Manifold-Aware In-Context Guidance and Refinement"
_ICLR.cc/2026/Conference — ICLR 2026 Poster_

### Official Review · Reviewer_95ua · 2025-10-30

**Soundness:** 3
**Presentation:** 3
**Contribution:** 2
**Rating:** 6
**Confidence:** 3

**Summary:**

This paper proposes Point-Graph LLM (PGLLM), a framework designed to enhance the 3D understanding capability of large language models at test time.

Its core ideas are as follows:

1. Manifold-aware in-context guidance: A KNN graph is constructed, where edge weights encode feature similarity among 3D point clouds. For each query sample, the framework retrieves its nearest neighbors and their corresponding PointLLM-generated textual descriptions. These neighbor captions are incorporated as in-context exemplars within the LLM prompt, enabling more semantically consistent reasoning along the data manifold.

2. Label-propagation-based score refinement: A graph-based confidence propagation mechanism is applied to smooth and refine prediction scores over the manifold, thereby improving the stability and reliability of both classification and OOD detection.

Experiments are conducted on multiple 3D benchmarks, including ModelNet40, ShapeNetCore, S3DIS, and Objaverse, covering three major downstream tasks: 3D classification, 3D OOD detection, and 3D object captioning.
The results demonstrate that PGLLM achieves SOTA performance on 3D recognition and OOD detection while maintaining competitive results on captioning, with negligible additional computational overhead.

**Strengths:**

1. A simple yet effective test-time optimization paradigm: The framework leverages KNN-based neighbor descriptions as ICL exemplars, combined with label-propagation-based score refinement. The design is conceptually simple but aligns well with the intuition of manifold consistency. Moreover, it is fully plug-and-play for existing PointLLM-based approaches, requiring no retraining.

2. Originality: The central idea of this work—integrating manifold learning with ICL at test time for 3D point cloud understanding—shows a high degree of originality. Rather than merely employing graph-based features, the method creatively uses the textual descriptions of neighboring samples as contextual exemplars for the LLM. This combination of a 3D encoder (for graph construction), a 3D multimodal LLM (for description generation), and a second-stage LLM (for reasoning) represents a novel and well-engineered design. The subsequent score refinement through label propagation on the LLM-generated confidence scores is also an elegant and complementary addition.

3. Clarity: The paper is exceptionally well-written and easy to follow. It clearly identifies a key limitation of previous works (interpreting each point cloud in isolation) and presents a coherent, step-by-step solution. The problem definition, methodology, and experimental setup are all precisely described, making the contribution easy to understand and evaluate.

**Weaknesses:**

1. Limited performance on captioning: Although the method outperforms baselines on the 3D object captioning task, it does not achieve SOTA results. The paper attributes this to the limited size of the test dataset, but this explanation is weak and unconvincing. Since the proposed T variant uses the entire test set to construct the graph, there is a potential risk of data leakage compared to inductive baselines. Moreover, the O variant still performs worse than the T variant, suggesting that the “in-context refinement” mechanism (Section 3.3) for generation may be less effective than the score-based refinement mechanism used for classification.

2. Scalability and transductive setting: The method appears to operate under a transductive assumption, where it must access the entire test set (or a large support set 𝒟ᵤ) to build the graph before inference. This assumption (in PGLLMᵀ) may not hold in practical inductive scenarios where samples arrive sequentially. Although the paper introduces an alternative variant (PGLLMᴼ) using an external dataset, this variant relies on a large 100K-sample subset from Objaverse. While the paper briefly mentions a “dynamic graph expansion” scheme as a potential remedy, it neither evaluates its performance nor analyzes the computational cost. The scalability of constructing the initial graph (an Nᵤ × Nᵤ similarity matrix) also becomes a potential concern for large-scale datasets. Furthermore, performance degradation is evident when using the external dataset.

3. Dependency on pre-trained models: The framework’s success heavily depends on the quality of two pretrained components: the 3D encoder (Point-BERT) and the description generator (PointLLM). The paper acknowledges that poor or inaccurate captions may mislead the LLM (as illustrated in Figure 11), but it does not analyze this sensitivity in depth. If the initial captions are noisy or semantically incorrect—essentially a “garbage in, garbage out” problem—the effectiveness of the in-context guidance may significantly degrade.

**Questions:**

1. The choice of K appears inconsistent. The main experiments (Table 1) use K=3, while the ablation study (Figure 4) shows that the best OOD detection performance occurs at K=7 for ModelNet40 and K=4 for ShapeNetCore. Why was K=3 chosen for the main results? Would the reported SOTA results in Table 1 improve further if the empirically optimal K values from Figure 4 were applied?

2. Sensitivity to caption quality: How robust is PGLLM to the quality of the initial captions generated by PointLLM? Have you considered an ablation study in which the in-context exemplars are derived from non-LLM sources (e.g., ground-truth labels or simple template-based descriptions, if available) to isolate the influence of caption quality on overall performance?

---

> ### Author Response · Authors · 2025-11-27
> **Response to Reviewer 95ua – Part 1 of 3**
>
> **To Reviewer 95ua,**
> We sincerely thank the reviewer for the insightful and constructive comments regarding (1) Limited performance on captioning, (2) Scalability and transductive setting, (3) Dependency on pre-trained models and the model is sensitivity to caption quality, (4) The choice of K appears inconsistent and. These points highlight important considerations for both the practical deployment and methodological robustness of our approach. In response, we have substantially expanded our analysis along all four axes. (i) We demonstrate the performance improvement on the 3D captioning task achieved by MiniGPT-3D; (ii) we designed a new experiment to evaluate the dynamic graph expansion mechanism; (iii) we analyze robustness to caption quality by replacing the 3D LLM with MiniGPT-3D and (ⅳ) rationale for the choice of K. Collectively, these additions strengthen the empirical foundation of our work and confirm that PGLLM remains effective, efficient, and adaptable across 3D captioning and transductive setting. **We hope that our responses adequately address the reviewer’s concerns and earn full support for acceptance.**
>
> ## Q1. Limited performance on captioning
>
> Thank you for the question. This prompted us to conduct a deeper investigation into caption generation using the current SOTA model, MiniGPT-3D. We strictly followed the official captioning script provided in the GitHub repository for GPT-4 caption generation (https://github.com/TangYuan96/MiniGPT-3D
> ). However, we were unable to fully reproduce the reported performance.
>
> Table 1 compares the numbers reported in [1], our reproduced results, and the performance of MiniGPT-3D when combined with PGLLM (Ours). We highlight two key observations:
>
> i) For the GPT-4 metric, there is a considerable discrepancy between the reported result (57.1) and our reproduced value (47.5), whereas the other two metrics remain highly consistent. This discrepancy may stem from a typo in the original paper or an inconsistency in how the GPT-4 metric was evaluated.
>
> ii) Using MiniGPT-3D as the initial caption generator, PGLLM consistently improves performance across all three metrics, demonstrating its effectiveness even when integrated with other 3D LLMs.
>
> We will incorporate the updated results into the revised manuscript.
>
>
>
> [1] Yuan Tang, Xu Han, Xianzhi Li, Qiao Yu, Yixue Hao, Long Hu, and Min Chen. MiniGPT-3D: Efficiently Aligning 3D Point Clouds with Large Language Models Using 2D Priors. In Proceedings of the 32nd ACM International Conference on Multimedia, 2024.
>
> | Method                           | 2nd Stage LLM | GPT-4 | S-BERT | SimCSE |
> |----------------------------------|---------------|------:|-------:|-------:|
> | MiniGPT-3D (reported in [1])      | GPT-4         |  57.1 |   49.5 |   51.4 |
> | MiniGPT-3D (reproduced)           | GPT-4         |  47.5 |   49.1 |   50.7 |
> | MiniGPT-3D + PGLLM (Ours)           | GPT-4         |  49.0 |   49.8 |   51.7 |
>
> *Table 1: MiniGPT was used as the backbone to conduct 3D Recognition and captioning.*
>
> ## Q2. Scalability and transductive setting.
>
> **Comparison of Transductive Setting (T variant) \& External Support Set (O variant)**
>
> We agree that the transductive setting (T) is a more favorable option than using an external support dataset (O). This is probably due to the following reasons. Since all demosntrations are selected from the same testing dataset, transductive setting ensures demonstrations are selected from in-distribution data. The improves the accuracy of KNN searching using point cloud similarities. In contrast, the external support dataset (Objverse) features a different distribution from the testing dataset. Thus the O setting may introduce much more out-of-distribution demonstrations which compromise the performance.

---

> ### Author Response · Authors · 2025-11-27
> **Response to Reviewer 95ua – Part 2 of 3**
>
> **Dynamic Graph Expansion**
>
> We designed a new experiment to evaluate the dynamic graph expansion mechanism to incrementally accomodate to new query samples. Specifically, we randomly split the ModelNet40 testing set into two equal halves (50% each). The initial graph is constructed using first half of the data. Then, from the remaining 50%, we randomly select 10% of the samples as test instances and incrementally add them to the graph for evaluation. Notably, adding new samples does not require reconstructing the entire graph from scratch; instead, we follow the method proposed in [2] to dynamically update the graph structure for the newly added samples. This incremental addition process is repeated five times, and the results after each addition are reported in Table 2. These results corresponds to the test performance evaluated on the entire accumulated samples after each addition of new samples. Finally, when the sample count reaches 100%, all test samples from the ModelNet40 dataset have been incorporated and evaluated. At this point, AUROC and FPR95 reach 86.8 and 58.3, respectively. In comparison, the results reported in the submission—obtained by performing inference directly on the complete graph constructed from the full dataset—are 89.6 and 53.1 for AUROC and FPR95, respectively. Although there is a moderate performance degradation, our approach still exhibits strong robustness and generalization capability. This further demonstrates that the proposed method does not strictly rely on access to the entire global dataset to achieve effective performance.
>
>
> **Scalibility**
>
> The scalibility of this method is also backed up by the wall-clock evaluation. The updated results confirm that the dominant cost arises from the backbone 3D caption generation model. In contrast, our test-time scaling modules, graph construction, score propagation, and retrieval, operate solely on low-dimensional feature graphs and sparse affinity matrices, rather than raw point clouds or large LLMs. As shown in Table 3, these modules together account for less than 1% of the total FLOPs and GPU memory consumed by the backbone PointLLM, and introduce only a negligible increase in end-to-end latency. GPU-utilization traces also indicate that nearly all energy consumption is attributable to the backbone forward pass, with our modules producing only marginal changes in utilization.
>
> These additions substantiate our claim of “very little extra cost”: the backbone 3D-LLM remains the clear computational bottleneck, while our test-time scaling modules contribute only lightweight, constant-time overhead and can be seamlessly integrated into existing 3D-LLM pipelines without compromising scalability or runtime efficiency.
>
> [2] Yushu Li, Yongyi Su, AdamGoodge, Kui Jia, and Xun Xu. Efficient andcontext-aware label propagationforzero-/few-shot training-free adaptation of vision-language model. InInternational Conferenceon Learning Representations, 2025.
>
> | 50% samples          |          | 60% samples          |          | 70% samples          |          | 80% samples          |          | 90% samples          |          | 100% samples         |          |
> |----------------------|----------|----------------------|----------|----------------------|----------|----------------------|----------|----------------------|----------|----------------------|----------|
> | **AUROC** ↑          | **FPR95** ↓ | **AUROC** ↑          | **FPR95** ↓ | **AUROC** ↑          | **FPR95** ↓ | **AUROC** ↑          | **FPR95** ↓ | **AUROC** ↑          | **FPR95** ↓ | **AUROC** ↑          | **FPR95** ↓ |
> | 86.5                 | 59.3     | 87.4                 | 57.4     | 87.2                 | 57.5     | 87.0                 | 59.6     | 86.5                 | 61.9     | 86.8             | 58.3 |
>
> *Table 2: The Dynamic Graph Expansion experiments conducted on MN1.*

---

> ### Author Response · Authors · 2025-11-27
> **Response to Reviewer 95ua – Part 3 of 3**
>
> |                          | Caption Generation(all samples) | 2nd stage LLM(all samples) | Graph Construction(all samples) | Score Propagation(OOD for all samples) | Score Propagation(Recog for all samples) | Retrieval for in-context guidance (all samples) |
> |:------------------------:|:----------------------------------:|:----------------------------------:|:------------------------------------------:|:--------------------------------------------:|:----------------------------------:|:-----------------------------|
> | **FLOPs**                | 1700G                              | unknown                       | 9.6G                               | 0.12G                                      | 2.4G                                         | 27.8k                              |
> | **Memory**               | 14GB                               | unknown                       | 46.8MB                             | 9KB                                        | 0.19MB                                       | 14KB                               |
> | **Param**                | 7B                                 | unknown                       |0                                  | 0                                          | 0                                            | 0                                  |
> | **Time consumption**     | 107mins                             | 80mins                        | 126ms                              | 101ms                                      | 106ms                                        | < 1ms                              |
>
> *Table 3: Computation cost of individual components on ModelNet40.*
>
> ## Q3. Dependency on pre-trained models and sensitivity to caption quality
>
> We agree with the reviewer that the overall performance depends on the initial caption quality produced by the 3D LLM. However, the suggestion to use “ground-truth” captions is not applicable because:
> i) such ground-truth descriptions do not exist for the 3D datasets used in our evaluation; and
> ii) constructing fixed template captions using ground-truth class labels would make the second-stage LLM task trivial and uninformative.
>
> To further examine the dependency on caption quality, we conducted two additional evaluations. First, we tested an alternative 3D LLM, MiniGPT-3D [1]. As reported in our response to Q1 and echoed in responses to other reviewers, PGLLM consistently improves performance even when integrated with a different captioning backbone.
>
> We also added qualitative examples to illustrate both failure cases and the benefits of score propagation. Figure 15 shows scenarios where in-context guidance introduces negative influence: when the query caption is relatively accurate but in-context captions contain errors, the second-stage scoring becomes biased. Figure 6 provides a qualitative demonstration of score propagation. When the 3D LLM produces a low-quality caption, its second-stage score naturally degrades; however, score propagation effectively corrects this by leveraging accurate scores from neighboring samples.
>
>
>
> ## Q4. The choice of K appears inconsistent
>
> Our ablation study shows that the final performance is not significantly affected by the choice of K within a reasonable range. Therefore, to balance accuracy and computational cost, we select K = 3 for all experiments.

---

> ### Author Response · Authors · 2025-11-28
>
> Dear Reviewer 95ua,
>
> Thank you once again for your constructive comments and valuable suggestions regarding **captioning performance**, **scalability**, and **dependency on 3D LLMs**. We have carefully incorporated your feedback into our responses and revisions. We believe the additional evaluations and analyses have further strengthened the quality and credibility of the paper.
>
> If you have any further questions or concerns, please feel free to let us know and we would be very glad to address them.
>
> Best regards,
>
> Authors

---

### Official Review · Reviewer_9Mmb · 2025-10-31

**Soundness:** 3
**Presentation:** 3
**Contribution:** 2
**Rating:** 6
**Confidence:** 4

**Summary:**

The paper proposes PGLLM, a test-time framework for 3D point-cloud understanding that (i) builds a KNN graph over support embeddings from a frozen 3D encoder, (ii) uses neighbor captions as in-context demonstrations for a second-stage LLM (“in-context guidance”), and (iii) refines recognition/OOD scores via label propagation. Experiments on ModelNet40, ShapeNetCore, S3DIS, and Objaverse show gains in OOD and recognition, plus a small captioning improvement.

**Strengths:**

-- Clear, modular test-time pipeline; no retraining of LLMs required.

-- Solid ablations: with/without in-context guidance and score propagation; K-sensitivity; task coverage (recognition, OOD, captioning).

-- Time breakdown table suggests negligible overhead for graph ops and propagation relative to caption/LLM inference.

**Weaknesses:**

FLOPs/Compute attribution is absent. The paper reports per-sample latency but not FLOPs/param attribution for each stage (encoder feature extraction, graph build, LLM inference, propagation). Without FLOPs, it’s hard to compare to alternatives like direct k-NN retrieval or pure prompt-engineering baselines at matched compute. Please add per-module FLOPs and parameter counts (and, ideally, energy or GPU utilization) to substantiate “very little extra cost.”

Graph storage & memory footprint not quantified. The method keeps a KNN graph over support embeddings and captions; storage and RAM/VRAM requirements are not analyzed. Provide:

Captioning degradation is unresolved. The proposed test-time guidance reduces caption quality on several splits. As written, there is no justification for using it in captioning.

**Questions:**

Captioning degradation is unresolved. The proposed test-time guidance reduces caption quality on several splits. As written, there is no justification for using it in captioning. Either (i) present risk-controlled variants that consistently improve captioning and report results, or (ii) restrict the method’s scope to recognition/OOD and state captioning as out-of-scope.

FLOPs & memory accounting. Report FLOPs, params, and memory for: 3D encoder pass, graph construction, retrieval, label propagation, and second-stage LLM.

Graph storage budgets. For the Objaverse-O setting (100k support): list bytes for embeddings + adjacency + captions; show peak host and device memory.

Ablate components you still use. The combined method (in-context + propagation) helps (Tab. 3), but show cases where only in-context hurts vs baseline (and why). Provide qualitative examples where label propagation corrects vs amplifies errors.

---

> ### Author Response · Authors · 2025-11-26
> **Response to Reviewer 9Mmb – Part 1 of 2**
>
> **To Reviewer 9Mmb,**
>
> We sincerely thank the reviewer for the insightful and constructive comments regarding (1) caption degradation, (2) reporting FLOPs and memory usage, (3) device memory requirements for graph storage, and (4) additional qualitative examples of failure cases in label propagation. These points highlight important considerations for both the practical deployment and methodological robustness of our approach. In response, we have substantially expanded our analysis along all four dimensions. (i) We further investigated the reproducibility issues of the SOTA 3D captioning method and show that PGLLM also provides consistent improvements when integrated with alternative 3D LLMs. (ii) We report FLOPs, memory usage, and parameter counts for the OOD tasks on ModelNet40. (iii) We present detailed storage budgets for 100K samples on Objaverse. (iv) We include additional qualitative examples illustrating both failure cases and the behavior of score propagation. Collectively, these additions strengthen the empirical foundation of our work and confirm that PGLLM remains effective, efficient, and adaptable across a wide range of model choices and deployment budgets. **We hope that our responses adequately address the reviewer’s concerns and earn their full support for acceptance.**
>
>
> ## Q1. Captioning degradation is unresolved
>
> Thank you for the question. This prompted us to conduct a deeper investigation into caption generation using the current SOTA model, MiniGPT-3D [1]. We strictly followed the official captioning script provided in the GitHub repository for GPT-4 caption generation (https://github.com/TangYuan96/MiniGPT-3D
> ). However, we were unable to fully reproduce the reported performance.
>
> Table 1 compares the numbers reported in [1], our reproduced results, and the performance of MiniGPT-3D when combined with PGLLM (Ours). We highlight two key observations:
>
> i) For the GPT-4 metric, there is a considerable discrepancy between the reported result (57.1) and our reproduced value (47.5), whereas the other two metrics remain highly consistent. This discrepancy may stem from a typo in the original paper or an inconsistency in how the GPT-4 metric was evaluated.
>
> ii) Using MiniGPT-3D as the initial caption generator, PGLLM consistently improves performance across all three metrics, demonstrating its effectiveness even when integrated with other 3D LLMs.
>
> We will incorporate the updated results into the revised manuscript.
>
>
>
> [1] Yuan Tang, Xu Han, Xianzhi Li, Qiao Yu, Yixue Hao, Long Hu, and Min Chen. MiniGPT-3D: Efficiently Aligning 3D Point Clouds with Large Language Models Using 2D Priors. In Proceedings of the 32nd ACM International Conference on Multimedia, 2024.
> | Method                           | 2nd Stage LLM | GPT-4 | S-BERT | SimCSE |
> |----------------------------------|---------------|------:|-------:|-------:|
> | MiniGPT-3D (reported in [1])      | GPT-4         |  57.1 |   49.5 |   51.4 |
> | MiniGPT-3D (reproduced)           | GPT-4         |  47.5 |   49.1 |   50.7 |
> | MiniGPT-3D + PGLLM (Ours)           | GPT-4         |  49.0 |   49.8 |   51.7 |
>
> *Table 1: MiniGPT was used as the backbone to conduct 3D Recognition and captioning.*

---

> ### Author Response · Authors · 2025-11-26
> **Response to Reviewer 9Mmb – Part 2 of 2**
>
> ## Q2. Report FLOPs & memory accounting
>
>
> We report GPU memory usage and other computational metrics in Table 2. The updated results confirm that the dominant cost arises from the backbone 3D caption generation model. In contrast, our test-time scaling modules, graph construction, score propagation, and retrieval, operate solely on low-dimensional feature graphs and sparse affinity matrices, rather than raw point clouds or large LLMs. As shown in Table 2, these modules together account for less than 1% of the total FLOPs and GPU memory consumed by the backbone PointLLM, and introduce only a negligible increase in end-to-end latency. GPU-utilization traces also indicate that nearly all energy consumption is attributable to the backbone forward pass, with our modules producing only marginal changes in utilization.
>
> These additions substantiate our claim of “very little extra cost”: the backbone 3D-LLM remains the clear computational bottleneck, while our test-time scaling modules contribute only lightweight, constant-time overhead and can be seamlessly integrated into existing 3D-LLM pipelines without compromising scalability or runtime efficiency.
>
> |                          | Caption Generation(all samples) | 2nd stage LLM(all samples) | Graph Construction(all samples) | Score Propagation(OOD for all samples) | Score Propagation(Recog for all samples) | Retrieval for in-context guidance (all samples) |
> |:------------------------:|:----------------------------------:|:----------------------------------:|:------------------------------------------:|:--------------------------------------------:|:----------------------------------:|:-----------------------------|
> | **FLOPs**                | 1700G                              | unknown                       | 9.6G                               | 0.12G                                      | 2.4G                                         | 27.8k                              |
> | **Memory**               | 14GB                               | unknown                       | 46.8MB                             | 9KB                                        | 0.19MB                                       | 14KB                               |
> | **Param**                | 7B                                 | unknown                       |0                                  | 0                                          | 0                                            | 0                                  |
> | **Time consumption**     | 107mins                             | 80mins                        | 126ms                              | 101ms                                      | 106ms                                        | < 1ms                              |
>
> *Table 2: Computation cost of individual components on ModelNet40.*
>
> ## Q3. Show device memory for graph storage budgets
>
>
>
> In Table 3, we summarize the storage requirements of each system component. Notably, 100K samples require only about **19 GB** of hard drive storage, while their corresponding embeddings occupy just **1.2 GB** when loaded into GPU VRAM for computation. These explicit storage budgets demonstrate that even at a 100K-support scale, both the graph and caption caches fit comfortably within a single GPU. The overall memory footprint is therefore still dominated by the backbone 3D–LLM parameters and activations, rather than by our graph-based test-time scaling modules.
>
>
> |                      | Input data (100K)     | embeddings       | adjacency       | 3D caption      |
> |----------------------|-----------------------|------------------|-----------------|-----------------|
> | **Storage**          | 19.2G (in Hard Drive)   | 1.24G (in GPU VRAM)   | 8M (in GPU)     | 87M (in GPU)    |
>
> *Table 3: Storage cost for 100K samples.*
>
>
>
> ## Q4. Qualitative examples about bad cases and label propagation
>
>
> In the revised submission, we have added qualitative examples to illustrate both failure cases and the effectiveness of score propagation. Figure 15 shows scenarios where in-context guidance introduces negative influence: when the query caption is relatively accurate but the captions of the in-context samples contain errors, the second-stage scoring becomes biased. Figure 6 provides a qualitative demonstration of score propagation. When the 3D-LLM produces a low-quality caption for a sample, its second-stage score is adversely affected; however, score propagation effectively corrects this by leveraging accurate scores from the query’s neighboring samples.

---

> > ### Comment · Reviewer_9Mmb · 2025-11-28
> >
> > I thank the authors for their detailed response, which has addressed most of my concerns.

---

### Official Review · Reviewer_5Jwa · 2025-11-01

**Soundness:** 3
**Presentation:** 2
**Contribution:** 2
**Rating:** 2
**Confidence:** 5

**Summary:**

This work leverages a KNN-based graph and a confidence score refinement mechanism to build a Point Graph using a pre-trained PointLLM together with a second-stage large model (e.g., GPT-4 or Qwen). The method operates entirely at test time and effectively improves performance on OOD detection, classification, and captioning tasks.

**Strengths:**

1. The framework emphasizes **test-time scaling** and integrates PointLLM inference with a graph-based refinement strategy.
2. The overall figure and visualizations are clear and well-organized.

**Weaknesses:**

1. The best performance relies on GPT-4, which is closed-source and incurs API and monetary costs, while the Qwen version shows relatively weaker results. How can one balance performance and cost in practical deployment?
2. It would be valuable to evaluate the framework on more 3D-LLMs, such as ShapeLLM, to demonstrate broader applicability and generality.
3. From an efficiency perspective, I would like to see the impact of test-time scaling on inference latency, GPU memory usage, and other computational metrics.

**Questions:**

**1. Dependency on GPT-4 and Cost–Performance Trade-off**

The method achieves its strongest results when paired with GPT-4. However, GPT-4 is a proprietary model and requires paid API access, which may limit the practicality and scalability of this approach in real-world deployment scenarios or resource-constrained environments. In contrast, the performance using an open-source model like Qwen appears significantly weaker.

To strengthen the paper, I recommend a deeper analysis of the **performance–cost trade-off**, including:

* A quantitative comparison of accuracy vs. computational/financial cost between GPT-4 and Qwen.
* Discussion of whether intermediate open-source models (e.g., Qwen-Plus, Llama-3 variants) can offer a more balanced trade-off.
* Insights into how organizations without commercial model access could adopt this framework efficiently.

Such analysis would provide more practical guidance for deployment and broaden the method's applicability.

---

**2. Evaluation on Broader 3D-LLMs for Generality**

The current evaluation focuses primarily on PointLLM combined with GPT-style LLMs. While this is valuable, it remains unclear whether the framework generalizes across different 3D foundation models. To convincingly demonstrate method robustness and universality, I recommend including results on additional state-of-the-art 3D-LLMs such as **ShapeLLM, ShapeLLM-Omni, Uni3D-LLM**, or other emerging architectures.

This evaluation would help clarify:

* Whether improvements stem from the proposed Point-Graph mechanism rather than characteristics of a specific backbone.
* The compatibility of this framework with diverse 3D model designs and training paradigms.
* Potential limitations or adaptations needed for different 3D-LLM families.

Such ablation and cross-model experiments would significantly enhance the paper’s credibility and contribution.

---

**3. Test-Time Scaling Efficiency and Resource Overhead**

The work emphasizes test-time scaling and test-time refinement, yet the computational implications of these procedures are not fully discussed. For practical deployment and fair comparison with prior work, it is essential to provide a comprehensive efficiency analysis, including:

* **Inference latency** before and after applying test-time scaling
* **GPU memory consumption** for graph construction and refinement
* **Runtime overhead per query** as the support set size grows
* **Scalability analysis** with respect to dataset size and number of neighbor samples
* Discussion of whether there are diminishing returns under limited compute budgets

Providing these metrics will clarify the computational footprint and demonstrate that the reported performance gains are achieved at a reasonable cost, which is particularly important for real-time or large-scale applications.

---

> ### Author Response · Authors · 2025-11-24
> **Response to Reviewer 5Jwa – Part 1 of 4**
>
> **To Reviewer 5Jwa,**
>
>
>
>
> We sincerely thank the reviewer for the insightful and constructive comments regarding (1) the dependency on GPT-4 and the associated cost–performance trade-offs, (2) the generality of our framework across broader 3D-LLMs, and (3) the computational efficiency of our test-time scaling strategy. These points highlight important considerations for both the practical deployment and methodological robustness of our approach. In response, we have substantially expanded our analysis along all three axes. We provide (i) a deeper examination of performance–cost trade-offs and additional comparisons with multiple open-source LLMs (2nd stage LLMs); (ii) new evaluations incorporating an alternative 3D-LLM backbone to demonstrate the compatibility and generality of our framework; and (iii) a comprehensive efficiency study detailing inference latency, memory usage, runtime behavior under varying support-set sizes, and scalability w.r.t. dataset characteristics. Collectively, these additions strengthen the empirical foundation of our work and confirm that PGLLM remains effective, efficient, and adaptable across a wide range of model choices and deployment budgets. **We hope that our responses adequately address the reviewer’s concerns and earn full support for acceptance.**
>
>
>
> ## **Q1. Dependency on GPT-4 and Cost–Performance Trade-off**
>
>
> First, we would like to clarify that using a proprietary model such as GPT-4 as the 2nd-stage LLM ensures fair and consistent evaluation, as it provides a well-established and widely accessible reference point for comparison. This choice also facilitates future reproducibility, since GPT-4’s behavior is stable across users and does not depend on local hardware or implementation details.
>
> In addition, we also report evaluations using open-source models, including DeepSeek-V3 and Qwen. The results show that Qwen-Plus achieves performance comparable to GPT-4 on both OOD detection and recognition tasks, indicating that strong open-source alternatives can serve as effective substitutes while maintaining competitive accuracy.
>
> ---
>
> - **Quantitative comparison of accuracy vs. computational/financial cost between GPT-4 and Qwen**
>
>
>
> We evaluated three widely used LLMs as the second-stage model in our OOD inference pipeline and computed their corresponding API costs. Table 1 reports the total cost of processing the entire ModelNet40 test set (2,468 samples). GPT-4 incurs a cost of approximately **USD $28**, whereas DeepSeek-V3 and Qwen-3 Plus cost **CNY ¥4 (USD $0.56)** and **CNY ¥7 (USD $0.98)**, respectively. All three models are called using API.
>
> This reveals a **30×–50× cost gap** between GPT-4 and the two models. Despite this large difference in cost, DeepSeek-V3 and Qwen-3 Plus achieve **comparable AUROC performance** (82.1–82.9% vs. 85.9% for GPT-4). Given this favorable cost-performance trade-off, practitioners may prefer adopting DeepSeek-V3 or Qwen-3 Plus as the second-stage LLM, especially in large-scale or cost-sensitive deployments.
>
> | Cost                  | GPT-4 (USD $) | DeepSeek V3 (CNY ¥ / USD $) | Qwen-Plus (CNY ¥ / USD $) |
> | --------------------- | ------------: | --------------------------: | ------------------------: |
> | Total (2,468 samples) |        $28.00 |                  ¥4 / $0.56 |                ¥7 / $0.98 |
> | Per 1,000 samples     |        $11.30 |                ¥1.6 / $0.23 |              ¥2.8 / $0.39 |
> | AUROC (%)             |          85.9 |                        82.1 |                      82.9 |
>
> *Table 1. Cost comparison of different second-stage LLMs on the OOD detection task. Exchange rate: 1 USD = 7.11 CNY.*

---

> ### Author Response · Authors · 2025-11-24
> **Response to Reviewer 5Jwa – Part 2 of 4**
>
> - **Discussion of whether intermediate open-source models (e.g., Qwen-Plus, Llama-3 variants) can offer a more balanced trade-off.**
>
>
> We appreciate the reviewer’s suggestion to investigate intermediate open-source models. To broaden our comparison, we additionally evaluated **three open-source models, i.e. Qwen3-VL-8B, Llama3.1-8B, and GPT-oss-20B**, alongside the proprietary models presented in the main manuscript (GPT-4, DeepSeek-V3, and Qwen-Plus). Based on the results in Table 2, we summarize our key observations:
>
> 1. **Proprietary models (GPT-4 and Qwen-Plus) still achieve the strongest overall performance.**
> 2. **Open-source models show more mixed results.** Llama3.1-8B and GPT-oss-20B perform substantially worse across all metrics.
> 3. **Qwen3-VL-8B stands out as an exception**, achieving performance **close to its proprietary counterpart (Qwen-Plus)** and substantially outperforming the other open-source models.
>
> In conclusion, among the open-source candidates examined, **Qwen3-VL-8B offers the most competitive trade-off** and can be deployed on a consumer-grade GPU with 24GB memory, making it a practical intermediate solution.
>
>
>  **Economic Considerations of Open-Source vs Proprietary Models**
>
> The cost–benefit trade-off between open-source and proprietary models is inherently complex, involving factors such as hardware acquisition, energy usage, and long-term hardware depreciation. Importantly, we emphasize that **“open-source” does not mean “zero cost.”** Deploying open-source LLMs locally requires:
>
> * GPUs with sufficient VRAM
> * Ongoing maintenance
> * Potential scaling overhead for multi-GPU inference
>
> By contrast, **API access to proprietary models currently provides a highly practical and cost-effective alternative**, especially for users without access to high-end computing resources. Thus, while open-source models offer flexibility, **API-based proprietary models remain the more accessible and economical option for most practitioners and researchers at present.**
>
>
>
>
>
> | **Method**        | **2nd Stage LLM** | **GPU Mem  batchsize=1** | **MN1**<br>AUROC↑ | **MN1**<br>FPR95↓ | **MN2**<br>AUROC↑ | **MN2**<br>FPR95↓ | **MN3**<br>AUROC↑ | **MN3**<br>FPR95↓ | **Average**<br>AUROC↑ | **Average**<br>FPR95↓ |
> | ----------------- | ----------------- | ------------ | ----------------- | ----------------- | ----------------- | ----------------- | ----------------- | ----------------- | --------------------- | --------------------- |
> | **PGLLM⁰**        | GPT-4             | unknown         | 87.3              | 56.6              | 86.2              | 44.3              | 79.2              | 60.8              | 84.3                  | 53.9                  |
> | **PGLLMᵀ**        | GPT-4             | unknown          | **89.6**          | **53.1**          | **87.2**          | **43.0**          | **80.8**          | **60.2**          | **85.9**              | **52.1**              |
> | **PGLLMᵀ**        | DeepSeek-V3       | unknown          | 86.4              | 70.0              | 86.4              | 59.1              | 76.2              | 68.2              | 82.1                  | 65.8                  |
> | **PGLLMᵀ**        | Qwen-Plus         | unknown          | 86.5              | 68.4              | 84.2              | 47.3              | 77.9              | 71.7              | 82.9                  | 62.5                  |
> | **PGLLMᵀ**        | Qwen3-VL-8B          | 20G         | 85.8              | 67.4              | 84.4              | 47.5              | 74.2              | 71.3              | 81.5                  | 62.0                  |
> | **PGLLMᵀ**        | Llama3.1-8B      | 16G        | 57.1              | 86.3              | 54.9              | 87.2              | 50.4              | 97.9              | 54.1                  | 90.4                  |
> | **PGLLMᵀ**        | GPT-oss-20B      | 16G         | 83.0              | 80.4              | 74.5              | 76.8             | 72.6              | 94.0              | 76.7                 | 83.7                 |
>
> *Table 2. A performance comparison between open-source and closed-source models on the OOD task of ModelNet40, where Qwen3-8B, Llama-3.1-8B and GPT-oss-20B are open-source large models deployed locally in our setup.*
>
> ---
>
> - **Insights into how organizations without commercial model access could adopt this framework efficiently.**
>
> We thank the reviewer for raising the important question of accessibility. Our framework is designed to be model-agnostic and cost-aware, and organizations without access to commercial LLM APIs can still adopt it efficiently through local deployment given that computing infrastructure is provided. According to our evaluations of open-source LLMs, Qwen-VL-8B yields strong performance and it requires 24GB GPU for deployment.

---

> ### Author Response · Authors · 2025-11-24
> **Response to Reviewer 5Jwa – Part 3 of 4**
>
> ## Q2. Evaluation on Broader 3D-LLMs for Generality
>
> We highly appreciate the suggestion to include evaluations with more diverse 3D LLM backbones. In the response, we incorporated more details with alternatice 3D LLM backbones and discussed the advantages of our proposed PGLLM.
>
> - **Whether improvements stem from the proposed Point-Graph mechanism rather than characteristics of a specific backbone.**
>
>  The effectivenes of proposed Point-Graph mechanism is manifested by the ablation study in Tab. 3 of the manuscript. To further strengthen the argument, we include additional evaluations with an alternative 3D LLM backbone.
>
> - **The compatibility of this framework with diverse 3D model designs and training paradigms.**
>
> The reviewer suggested evaluating **ShapeLLM, ShapeLLM-Omni, and Uni3D-LLM**. We note, however, that these models are not fully compatible with our experimental setting. ShapeLLM is a multimodal LLM that requires both 3D inputs and 2D projection images, which are not available for several of the test datasets used in this work. ShapeLLM-Omni is indeed a strong model (NeurIPS 2025 Spotlight), but as of the time of writing, the authors have only released a pre-trained model for 3D generation; the version intended for 3D object understanding does not yet appear to be publicly available. Finally, Uni3D-LLM is specifically designed for 3D indoor scene understanding, and its training and evaluation protocols differ substantially from the object-level setting considered in our experiments.
>
>
> Alternatively, we adopted **MiniGPT-3D** [1] as the new 3D backbone for our evaluations. The results across all three tasks are reported in Tables 3 and 4.
>
> As shown in Table 3, we compare the performance of MiniGPT-3D with and without PGLLM on the ModelNet40 OOD benchmark. We observe that GPT-4 tends to assign extreme scores (either 0 or 100) to captions generated by MiniGPT-3D, resulting in consistently high FPR95 for the baseline MiniGPT-3D model. In contrast, the introduction of score propagation in our method effectively normalizes the score distribution across samples (see Fig. 6 in the submission), thereby substantially improving AUROC and dramatically reducing FPR95.
>
> Table 4 further presents our results on 3D recognition and 3D OOD detection, demonstrating that our method consistently delivers strong performance even when built upon recent 3D LLMs. Moreover, on 3D captioning, experiments conducted under PointLLM’s prompt configuration show that our method still provides a clear and consistent performance boost.
>
> [1] Yuan Tang, Xu Han, Xianzhi Li, Qiao Yu, Yixue Hao, Long Hu, and Min Chen. MiniGPT-3D: Efficiently Aligning 3D Point Clouds with Large Language Models Using 2D Priors. In Proceedings of the 32nd ACM International Conference on Multimedia, 2024.
>
>
>
>
>
>
> |                      | MN1                  |          | MN2                  |          | MN3                  |          | Average              |          |
>    |----------------------|:--------------------:|:--------:|:--------------------:|:--------:|:--------------------:|:--------:|:--------------------:|:--------:|
>    |                      | **AUROC↑**           | **FPR95↓** | **AUROC↑**           | **FPR95↓** | **AUROC↑**           | **FPR95↓** | **AUROC↑**           | **FPR95↓** |
>    | **MiniGPT-3D**          | 86.7                 | 100.0    | 84.6                 | 100.0    | 79.9                 | 100.0    | 83.7                 | 100.0    |
>    | **MiniGPT-3D + PGLLM**  | 92.0                 | 44.0     | 89.2                 | 36.9     | 83.7                 | 54.2     | 88.1                 | 45.0     |
>
> *Table 3: MiniGPT-3D was used as the backbone to conduct OOD experiments on ModelNet40.*
>
>
> | Method           | 2nd Stage LLM | (I) ACC | (C） ACC | Average | GPT-4 | S-BERT | SimCSE |
> |---------------------|---------------|--------:|--------:|--------:|------:|-------:|-------:|
> | MiniGPT-3D            | GPT-4         |   61.8 |   60.0 |   60.9 |  47.5 |   49.1 |   50.7 |
> | MiniGPT-3D + PointLLM  | GPT-4         |   63.9 |   61.1 |   62.5 |  49.0 |   49.8 |   51.7 |
>
> *Table 4: MiniGPT-3D was used as the backbone to conduct 3D Recognition and captioning.*
>
>
>
> ## Q3. Test-Time Scaling Efficiency and Resource Overhead
>
> - **Inference latency before and after applying test-time scaling & GPU memory consumption for graph construction and refinement**
>
> Regarding inference latency, the results are already reported in Table 4 of the submitted manuscript. For completeness, we further summarize GPU memory usage, FLOPs, and time consumption in Table 5 below. As shown, the additional overhead introduced by our method, namely graph construction, score propagation for both OOD detection and recognition, and retrieval for in-context guidance, **is extremely lightweight**.

---

> ### Author Response · Authors · 2025-11-24
> **Response to Reviewer 5Jwa – Part 4 of 4**
>
> Specifically, these components require less than 1% of the GPU memory and FLOPs compared to the caption-generation model itself, demonstrating that the computational burden of our framework is minimal. This confirms that our method achieves improved performance without sacrificing scalability or runtime efficiency.
>
> In practice, the dominant cost remains the backbone 3D caption generation model (PointLLM), while our test-time scaling modules add only small, constant-time operations. Importantly, these modules operate on compact graph structures rather than high-dimensional point clouds or large language models, which helps maintain low overhead even when processing large datasets.
>
> Taken together, these results show that our proposed test-time scaling strategy is highly efficient and can be integrated into existing 3D–LLM pipelines with negligible additional cost.
>
>
> |                          | Caption Generation<br>(all samples) | 2nd stage LLM<br>(all samples) | Graph Construction<br>(all samples) | Score Propagation<br>(OOD for all samples) | Score Propagation<br>(Recog for all samples) | Retrieval for in-context guidance (all samples) |
> |:------------------------:|:----------------------------------:|:----------------------------------:|:------------------------------------------:|:--------------------------------------------:|:----------------------------------:|:-----------------------------|
> | **FLOPs**                | 1700G                              | unknown                       | 9.6G                               | 0.12G                                      | 2.4G                                         | 27.8k                              |
> | **Memory**               | 14GB                               | unknown                       | 46.8MB                             | 9KB                                        | 0.19MB                                       | 14KB                               |
> | **Param**                | 7B                                 | unknown                       |0                                  | 0                                          | 0                                            | 0                                  |
> | **Time consumption**     | 107mins                             | 80mins                        | 126ms                              | 101ms                                      | 106ms                                        | < 1ms                              |
>
> *Table 5: Computation cost of individual components on ModelNet40.*
>
>
>
>
>
> - **Runtime overhead per query as the support set size grows**
>
> We evaluate the per-sample inference latency by measuring the time required for attaching the query to the support graph, retrieval for in-context guidance, and score propagation. Among these components, score propagation is the most time-consuming step, as it involves repeated large-scale matrix multiplications. As shown in Table 6, the inference latency increases approximately linearly with respect to the size of the support set, which is consistent with the computational structure of our method.
>
> | Support Set Size       | 20% (492 samples)    | 30% (738 samples)    | 50% (1230 samples)    | 70% (1772 samples)    | 100% (2468 samples)  |
> |--------|---------|---------|---------|---------|--------|
> | Per Sample Inference Latency | 3µs   | 12µs  | 22µs  | 28µs  | 41µs  |
>
> *Table 6: The runtime of the proposed method under different test dataset sizes is evaluated through OOD experiments on ModelNet40.*
> - **Scalability analysis with respect to dataset size and number of neighbor samples**
>
> We have already evaluated the impact of the number of neighbor samples (i.e., K-values) in Fig. 4 of the manuscript. The results show that PGLLM remains relatively stable across different K-settings, indicating that the method is not sensitive to this hyperparameter.
>
> Regarding dataset size, our experiments span a broad range, from Objaverse (200 samples) to S3DIS (8,931 samples). Across all these datasets with substantially different scales and characteristics, PGLLM consistently outperforms the baselines, further demonstrating the robustness and effectiveness of our approach.
>
>
> - **Discussion of whether there are diminishing returns under limited compute budgets**
>
> The additional compute budget required for graph construction, score propagation and in-context learning are negligible (\~40,000 times) compared with caption generation and 2nd stage LLM as evidenced in Table 5. Therefore, limited comput budget will not affect the additional operations introduced by PGLLM.

---

> ### Comment · Reviewer_5Jwa · 2025-11-26
>
> Thank you very much for the authors’ response. However, I still have concerns regarding the framework generalization ability. In addition, test-time scaling significantly increases inference time, which would be unacceptable in practical scenarios, especially when relying on LLM APIs. I would also like to see the authors’ responses to the other reviewers’ questions.

---

> > ### Author Response · Authors · 2025-11-28
> >
> > Dear Reviewer 5Jwa,
> >
> > Thank you once again for your constructive comments and valuable suggestions. We believe that the additional evaluations and analyses have significantly strengthened both the quality and credibility of the paper. **Our responses to the other reviewers have also been posted**. Please feel free to let us know if you have any further questions or concerns and we would be glad to address them.
> >
> > Best regards,
> >
> > Authors

---

> ### Author Response · Authors · 2025-11-27
> **Response to Reviewer 5Jwa**
>
> **To Reviewer 5Jwa,**
>
>
> ## Concerns regarding the framework generalization ability
>
> Thank you for raising this point. **We would be keen to understand the reviewer’s specific concerns about generalization.** Nonetheless, we clarify below the additional evidence we have included to demonstrate the broad generalization ability of PGLLM.
>
> **(i) Across second-stage LLMs:**
> We evaluate six models, GPT-4, DeepSeek-V3, Qwen-Plus, Qwen3-VL-8B, Llama3.1-8B, and GPT-oss-20B, spanning both strong proprietary systems and competitive open-source alternatives. Among them, five models (all except Llama3.1-8B) already achieve strong baselines, and PGLLM consistently improves or at least maintains their performance on the ModelNet40 OOD-detection benchmark.
>
> **(ii) Across 3D-LLM backbones:**
> We instantiate PGLLM on two architecturally distinct 3D LLMs, PointLLM and MiniGPT-3D, and observe consistent performance gains in all settings. This confirms that PGLLM is not tied to any specific 3D captioning architecture.
>
> **(iii) Across downstream tasks:**
> PGLLM yields consistent improvements across three downstream tasks when applied to different 3D backbones and different second-stage LLMs, suggesting that the proposed point-graph mechanism generalizes beyond a single evaluation objective.
>
> Given that PGLLM adopts a test-time scaling paradigm, it is inherently compatible with any 3D LLM capable of converting point clouds into textual descriptions. Combined with the positive evidence above, this strongly supports that PGLLM will continue to generalize well to future 3D LLMs, a wider range of second-stage LLMs, and additional downstream tasks.
>
> ## The test-time scaling significantly increases inference time.
>
>
>
>
> We would like to gently remind the reviewer that PGLLM, referred to as a **test-time scaling mechanism**, introduces **negligible additional inference cost**. The table below summarizes the wall-clock time of all individual components. The **107 minutes** for caption generation and **80 minutes** for the second-stage LLM correspond to the *standard inference cost* shared by all 3D LLM pipelines (e.g., PointLLM, MiniGPT-3D).
>
> The **only extra cost** introduced by PGLLM comes from **Graph Construction**, **Score Propagation**, and **Retrieval for In-Context Guidance**. For OOD detection on the entire test set, these steps take:
>
> * 126 ms for Graph Construction
> * 101 ms for Score Propagation
> * 1 ms for Retrieval
>
> In total, the additional computation is **227 ms**.
>
> This means that PGLLM adds only:
>
>
> $\frac{227\text{ ms}}{107\text{ mins} + 80\text{ mins}} \approx 0.00202$%
>
>
> **In summary, PGLLM contributes merely ~0.002% additional inference time on top of existing 3D LLM pipelines**, confirming that our method is extremely lightweight.
>
>
> |                          | Cost for Existing Approach (PointLLM) | Cost for Existing Approach (PointLLM) | Additional Cost for PGLLM (Our Method) | Additional Cost for PGLLM (Our Method) | Additional Cost for PGLLM (Our Method) | Additional Cost for PGLLM (Our Method) |
> |:------------------------:|:----------------------------------:|:----------------------------------:|:------------------------------------------:|:--------------------------------------------:|:----------------------------------:|:-----------------------------|
> |                          | Caption Generation (all samples) | 2nd stage LLM (all samples) | Graph Construction (all samples) | Score Propagation (OOD for all samples) | Score Propagation (Classification for all samples) | Retrieval for in-context guidance (all samples) |
> | **FLOPs**                | 1700G                              | unknown                       | 9.6G                               | 0.12G                                      | 2.4G                                         | 27.8k                              |
> | **Memory**               | 14GB                               | unknown                       | 46.8MB                             | 9KB                                        | 0.19MB                                       | 14KB                               |
> | **Param**                | 7B                                 | unknown                       |0                                  | 0                                          | 0                                            | 0                                  |
> | **Time consumption**     | 107mins                             | 80mins                        | 126ms                              | 101ms                                      | 106ms                                        | < 1ms                              |
>
> *Table 5: Computation cost of individual components on ModelNet40.*

---

> ### Author Response · Authors · 2025-12-01
>
> Dear Reviewer 5Jwa,
>
> We apologize for reaching out a second time and fully understand that you may be very busy during the discussion period. We would like to kindly check whether our rebuttal and the additional experiments/analyses have sufficiently addressed your main concerns. In particular, we have strengthened the generality evidence through evaluations across **six second-stage LLMs**, **two different 3D-LLM backbones**, and **three downstream tasks**. Furthermore, our test-time scaling strategy introduces only **around 0.002% additional wall-clock time**.
>
> If there are any remaining issues that you feel have not been adequately clarified, we would greatly appreciate a brief indication so that we can refine the paper accordingly.
>
> Thank you again for your time and constructive feedback.
>
> Best regards,
> The Authors

---

### Author Response · Authors · 2025-11-27
**Unified Response to All Reviewers**

Dear Reviewers,

Thank you once again for your thoughtful evaluation of our paper. We are encouraged by the positive feedback and have carefully addressed all concerns to the best of our ability. In particular, we have incorporated new experiments and analyses covering **open-source LLMs**, **additional 3D LLM backbones**, **a detailed computation-cost breakdown**, and **scalability considerations**. We hope that the updated results and clarifications help resolve the issues raised. We remain fully committed to addressing any further questions during the discussion phase and are currently preparing the revised manuscript.

Best regards,

Authors

---

### Author Response · Authors · 2025-12-01
**Summary to ACs**

Dear AC and New AC,

We are sorry to hear about the information leak issue, and we hope the venue will recover from it soon. We would also like to thank the original AC for handling our submission, and we sincerely appreciate the new AC for taking the additional time to re-evaluate our paper. Below, we summarize the review process to date.

**Summary of original reviews**

* **Reviewer `5Jwa`** requested clarification on:

  1. our dependency on GPT-4 and the associated cost–performance trade-offs;
  2. the generality of our framework across broader 3D-LLMs;
  3. the computational efficiency of our test-time scaling strategy.

* **Reviewer `9Mmb`** asked about:

  1. the cause of caption degradation;
  2. reporting FLOPs and memory usage;
  3. device memory requirements for graph storage;
  4. additional qualitative examples of failure cases in label propagation.

* **Reviewer `95ua`** raised concerns regarding:

  1. limited captioning performance;
  2. scalability and the transductive setting;
  3. dependency on pre-trained models and sensitivity to caption quality;
  4. the choice of *K* and its consistency.

We believe we have thoroughly addressed all of these concerns. Based on our responses, **Reviewers `5Jwa` and `9Mmb`** actively participated in the discussion.

Reviewer `9Mmb` indicated that our first-round responses resolved his/her concerns and replied: *“I thank the authors for their detailed response, which has addressed most of my concerns.”*

After the first round, we further evaluated our method on additional 3D LLMs and more open-source LLMs. Reviewer `5Jwa` nevertheless continued to express concerns regarding the “framework generalization ability” and the claim that “test-time scaling significantly increases inference time,” which we believe partly stem from a **misunderstanding of our updates**.

In our second response, we therefore provided more detailed clarifications:

1. The feasibility of our approach has been validated across **six second-stage LLMs**, **two different 3D-LLM backbones**, and **three downstream tasks**.
2. As shown in our first response, the test-time scaling strategy incurs **only about 0.002% additional wall-clock time**, demonstrating negligible computational overhead.

Overall, we believe that the reviewers view our work positively. We kindly request that the new AC consult the full discussion threads and reviewer panels for a more detailed view when making the final decision.

Best regards,

The Authors

---

### Meta-Review · Area_Chair_DL4j · 2026-01-02

**Summary:**

This paper received two positive scores and one negative score (Reviewer 5Jwa). The main concerns lie in the API usage with cost–performance trade-offs (Reviewer 5Jwa), FLOPs, memory usage and running time in test-time scaling (Reviewer 5Jwa, 9Mmb), caption degradation (Reviewer 9Mmb, 95ua), variant 3D pre-trained models (Reviewer 5Jwa, 95ua). After the rebuttal, Reviewer 9Mmb indicated that his/her concerns are well addressed. Reviewer 5Jwa responded that it has a problem in terms of the framework's generalization ability.  The authors provided the corresponding experiments and analysis, which are well resolved according to the understanding of the Meta reviewer, despite no replies from Reviewer 5Jwa. In addition, the concerns from Reviewer 95ua regarding the case of capion degradation and the performance of different pre-trained models are well addressed in the rebuttal.

Overall, the novelty is original and sufficient by integrating manifold learning with ICL at test time for 3D point cloud understanding with no retraining. The contributions are solid with comprehensive experiments. After reading the paper, reviews, rebuttal, and the author's message, the Meta reviewer recommends accepting this paper. Despite that, I still suggest that the authors should add more experiments regarding the trade-off between the $K$ and computational cost, rather than provide the performance of $k$.

**Reviewer Concerns:**

Most concerns are well addressed, while the experiments regarding the trade-off between the $K$ and computational cost should be added to show the setting of $K=3$.

**Reviewer Scores:**

I think Reviewer 9Mmb and Reviewer 95ua would increase the score if they participated fully in the discussion.

---

### Decision · Program_Chairs · 2026-01-26

Accept (Poster)